# *ReGen*: Generative Robot Simulation via Inverse Design

**Phat Nguyen**[1], **Tsun-Hsuan Wang**[1], **Zhang-Wei Hong**[1], **Erfan Aasi**[1], **Andrew Silva**[2],
**Guy Rosman**[2], **Sertac Karaman**[1], **Daniela Rus**[1*]
[1]Massachusetts Institute of Technology      [2]Toyota Research Institute
{peterng,tsunw,zwhong,eaasi,sertac,rus}@mit.edu
{andrew.silva,guy.rosman}@tri.global

## ABSTRACT

Simulation plays a key role in scaling robot learning and validating policies, but constructing simulations remains a labor-intensive process. This paper introduces *ReGen*, a generative simulation framework that automates simulation design via inverse design. Given a robot's behavior—such as a motion trajectory or an objective function—and its textual description, *ReGen* infers plausible scenarios and environments that could have caused the behavior. *ReGen* leverages large language models to synthesize scenarios by expanding a directed graph that encodes cause-and-effect relationships, relevant entities, and their properties. This structured graph is then translated into a symbolic program, which configures and executes a robot simulation environment. Our framework supports (i) augmenting simulations based on ego-agent behaviors, (ii) controllable, counterfactual scenario generation, (iii) reasoning about agent cognition and mental states, and (iv) reasoning with distinct sensing modalities, such as braking due to faulty GPS signals. We demonstrate *ReGen* in autonomous driving and robot manipulation tasks, generating more diverse, complex simulated environments compared to existing simulations with high success rates, and enabling controllable generation for corner cases. This approach enhances the validation of robot policies and supports data or simulation augmentation, advancing scalable robot learning for improved generalization and robustness. Please check our website here: https://regen-sim.github.io/.

## 1 INTRODUCTION

Simulated environments play a vital role in validating robotic systems and provide platforms for robots to acquire complex skills, including autonomous driving (car, 2020; Amini et al., 2022; Gulino et al., 2024), manipulation (Zhu et al., 2020; James et al., 2020; Nasiriany et al., 2024a), and locomotion (Rudin et al., 2022; Makoviychuk et al., 2021). Unlike real-world learning or testing, simulations offer access to privileged states, enable unlimited exploration, and support large-scale parallel computation—all without the need for heavy investment in robotic hardware. Classical simulation methods often rely on manually crafted environments and predefined scenarios, which require significant human expertise and effort in both setup and maintenance Brockman (2016); Müller et al. (2018). These traditional approaches, while effective, are often limited in flexibility and scalability compared to the newer techniques. A more recent paradigm, *generative simulation*, leverages generative artificial intelligence (AI) to automate the creation of simulations, greatly reducing the human effort and tedious process typically involved. Promising progress has been made in asset generation (Wang et al., 2023a; Siddiqui et al., 2024), scene layout design (Höllein et al., 2023; Yang et al., 2024), and task and environment creation (Wang et al., 2024d; 2023c).

Generative simulation offers the potential to create infinite environments for robots to learn and be tested in. However, previous methods often encounter significant limitations in generating low-level control—such as a trajectory—from high-level textual descriptions, constraining the diversity

---

*This work is supported by Toyota Research Institute (TRI). It, however, reflects solely the opinions and conclusions of its authors and not TRI or any other Toyota entity.

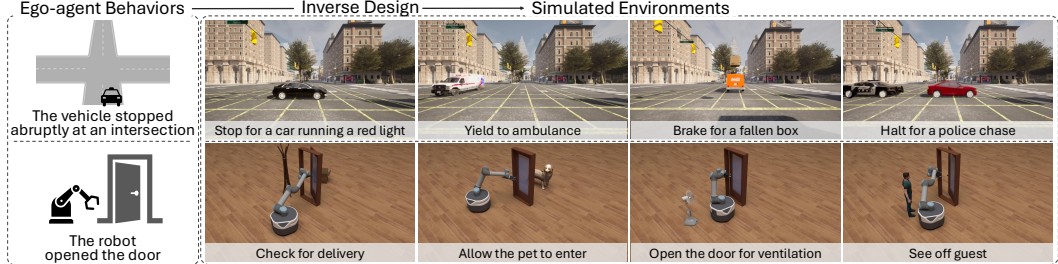

Figure 1: Given a robot behavior (such as a trajectory or an underlying objective function) and its textual description, **ReGen** generates simulated environments that could have caused the behavior.

and complexity of the robot simulations. For instance, these approaches often require training a new policy for each simulation based on a newly generated reward function, making this process the primary computational bottleneck. We leverage the insight that behaviors are relatively limited compared to the diverse environments in which they occur. For instance, the abrupt stopping of a self-driving car can apply to various contexts, such as a red traffic light, a pedestrian stepping into the road, or an approaching police car with its siren on (see Figure 1). To address this, we draw inspiration from *inverse design* (Molesky et al., 2018), a concept widely used in computational design that starts with desired properties or outcomes. For example, given a target flow pattern, the geometry of a fluidic device is optimized accordingly. In generative simulation, we propose generating or designing environments conditioned on the agent's behavior. The benefits of this framework can be investigated from different perspectives:

**A validation perspective.** Generative simulation via inverse design enables the conditional generation of simulated environments based on specific robot behaviors. This approach facilitates the unit testing of robotic systems, where the unit is defined at the behavior level[1]. By tailoring environments to specific behaviors, we can systematically evaluate how well a robot performs under a controlled set of diverse yet relevant contexts, rather than relying on random or uncontrolled scenarios.

**An augmentation perspective.** A natural extension from validation is to incorporate failed test cases into the robot's learning pipeline as a form of data augmentation. This approach strengthens the robustness of the robot's behavior across diverse and relevant environmental contexts. Additionally, this inverse approach can be seen as a method for augmenting existing simulated environments. Given a simulated environment and a robot policy exhibiting a relevant behavior, inverse design enables the sensible addition or removal of entities based on the context. This is achieved by reasoning backward from the robot's behavior to identify what elements in the environment are relevant.

In this work, we propose an inverse design approach for generative simulation. Given a robot's behavior—such as a motion trajectory or an objective function—along with their textual description, our method generates relevant simulated environments. We represent the environment as a directed graph, where nodes correspond to events, entities, or properties, while edges capture causal relationships (e.g., a car stopping "due to" an ambulance) or dependencies (e.g., a siren on an ambulance). This structured representation is then transformed into a symbolic program that configures and executes a robot simulation environment. In summary, we contribute:

- An inverse design approach for generative simulation demonstrated in autonomous driving and manipulation with abilities to (i) augment simulations based on ego-agent behaviors, (ii) generate controllable, counterfactual scenarios, (iii) reason about agent cognition and mental states, and (iv) handle distinct sensing modalities, such as braking due to faulty GPS signals.
- A method for synthesizing simulatable scenarios using LLM-guided graph search to construct a directed graph that encodes causal relationships, entities, and properties, which is then converted into an executable symbolic program for robot simulation.
- Extensive experiments that showcase greater diversity of generated environments compared to existing simulations, controllable generation of corner cases for safety-critical applications like driving, and superior complexity of generated environments, which produces vision-language-action datasets that are more challenging to vision language models (VLMs) than existing datasets.

---

[1]The level of granularity at which the behavior should be defined is beyond the scope of this paper and remains an open research question for future exploration.

---

**Algorithm 1** Graph Expansion

---

**Input:** $v_{\text{behavior}} \in \mathcal{V}_{\text{event}}$    **Output:** $G \in \mathcal{G}$
**procedure** POTENTIALLY_CONNECT_TO_NODE($G, v, \ldots$)
     $v_{\text{candidate}} = \texttt{node\_proposal}(v, \ldots)$
     G, edge_added $= \texttt{edge\_construction}(v, v_{\text{candidate}}, G)$
     **return** G, $v_{\text{candidate}}$, edge_added
**end procedure**
**Initialize:** $G = \{\} \in \mathcal{G}$;   $G \leftarrow v_{\text{behavior}}$
**while** $\exists \texttt{ input\_degree}(v \in G) < 1$ **do**
     $v_{\text{event}} \sim \{v \mid \texttt{input\_degree}(v \in G) < 1\}$
     G, $v'_{\text{event}}$, _ = $\texttt{potentially\_connect\_to\_node}(G, v_{\text{event}}, \ldots)$
     break whenever by the user.
**end while**
**for** $\{v_{\text{event}} \mid v \in G, \; v \in \mathcal{V}_{\text{event}}\}$ **do**
     G, $v_{\text{entity}}$, edge_added = $\texttt{potentially\_connect\_to\_node}(G, v_{\text{event}}, \ldots)$
     **if** edge_added **then**
         **for** All possible property types **do**
             G, $v_{\text{property}}$, _ = $\texttt{potentially\_connect\_to\_node}(G, v_{\text{entity}}, \ldots)$
         **end for**
     **end if**
**end for**

---

## 2 METHOD

We present **ReGen**, a framework that takes as inputs a behavior $B = \{\tau, \mathcal{R}\}$, where $\tau$ is a motion trajectory—generated using motion primitives—and a reward function $\mathcal{R}$, along with its textual description and a graph-based representation of the simulator database $\mathcal{D}_{\text{asset}}$, to generate plausible simulations. For example, given "vehicle changed lanes," our method simulates scenarios such as *yielding to an ambulance*, *overtaking a slow truck*, and *merging into an open lane*. We assume access to a simulation engine, such as CARLA (car, 2020) for autonomous driving or PyBullet (Coumans & Bai, 2016–2021) for manipulation, along with a database $\mathcal{D}_{\text{asset}}$. For further details on implementation, see (Appendix A.1).

At a high level, the method begins by synthesizing plausible scenarios through graph search using an LLM, where a directed graph is iteratively expanded by introducing new plausible contexts and fine-grained details, such as entities and their properties (Section 2.1). Next, for a scenario generated through the graph, we leverage an LLM to generate a symbolic program that formulates high-level constraints, serving as a verifier to ground the scenario in simulation (Section 2.2).

### 2.1 INVERSE DESIGN VIA GRAPH EXPANSION

The inverse design process in Algorithm 1 begins by representing the input behavior description as a leaf node. The graph $\mathcal{G}$ is then expanded *backward* through two atomic steps: *node proposal* and *edge construction*, applied to different types of nodes and edges. Both steps employs the gpt-4o-2024-08-06 model with parameters temp=0 and top-p=0. A key perspective is that graph expansion systematically explores the LLM's knowledge space, uncovering and organizing latent information into a structured graph.

**Node Proposal** is the process of generating candidate nodes $\mathcal{V}_{\text{candidate}} = \{\mathcal{V}_{\text{event}}, \mathcal{V}_{\text{entity}}, \mathcal{V}_{\text{property}}\}$ that may connect to the existing graph via a source node $v_{\text{source}}$. We consider three types of nodes: event, entity, and property nodes. First, event nodes $v_{\text{event}}$ represent causal variables (e.g., an event "yielding to an ambulance" is a cause of another event "ego-vehicle stopping"). We use an LLM to propose candidate event nodes LLM $(v \in \mathcal{V}_{\text{event}} | v_{\text{source}}, \texttt{prior})$, treating the source node as the effect. The prior is an optional natural language description that injects additional context and can be sourced from various inputs, such as user preferences or another LLM. Next, entity nodes $v_{\text{entity}}$ represent static objects (e.g, debris) or dynamic actors (e.g., an ambulance) and are proposed from a fixed set of supported assets $\mathcal{D}_{\text{asset}}$ in the simulation engine $v \in \mathcal{V}_{\text{entity}} \subseteq \mathcal{D}_{\text{asset}}$. Lastly, property nodes $v_{\text{property}}$ define attributes for each entity, including elements proposed by an LLM, such as

location (e.g. "in front of the ego"), or possible states retrieved from $\mathcal{D}_{\text{asset}}$. For example, candidate property nodes for a traffic light represent all its possible states in the simulator, such as "red," "green," "yellow", and "off". These proposed nodes serve as candidates for the edge construction step. See Appendix A.1.2 for examples and full prompts in Appendix A.5.1.

**Edge Construction** determines which candidate nodes should be connected to a source node $v_{\text{source}}$ by evaluating possible connections simultaneously, rather than pairwise (Jiralerspong et al., 2024). Formally, it involves mapping $\text{LLM} : (v_{\text{source}}, \mathcal{V}_{\text{candidate}}) \rightarrow \{\text{True}, \text{False}\}$, where $\mathcal{V}_{\text{candidate}} = \{v_1, v_2, \ldots, v_n\}$ represent the sets of candidate nodes from the node proposal step. Edges are then constructed for all $\{v_i \in \mathcal{V}_{\text{candidate}} : \text{LLM}(v_{\text{source}}, v_i) = \text{True}\}$. This mapping leverages an LLM as a general classifier to evaluate the plausibility of each candidate connection in the context of the source node. Our graph expansion considers three types of edge constructions: event-to-event, entity-to-event, and property-to-entity, with the order indicating the direction. For event-to-event edges, the LLM validates direct causal relationships using common-sense reasoning—for example, an emergency vehicle behind the ego-vehicle may cause the ego to pull over, while one ahead may not. For entity-to-event edges, the LLM selects relevant entities, while for property-to-entity edges, connections are determined by supported properties relevant to the entity's role. The edge construction process performs rejection sampling by discarding implausible connections and ensuring only edges corresponding to an available entity or property in $\mathcal{D}_{\text{asset}}$ are added to the graph. See Appendix A.1.3 for examples and Appendix A.5.2 for full prompts.

**Graph Expansion** is an iterative process that augments the graph through successive node proposals and edge construction. The process begins with the initialization of the graph $\mathcal{G} = (\mathcal{V}, \mathcal{E})$, where the initial sets of nodes $\mathcal{V} = \{\mathcal{L}_B\}$ contains the input behavior description $\mathcal{L}_B \in \mathcal{V}_{\text{event}}$. In the node proposal stage, candidate event nodes are generated for all event nodes that lack an incoming edge, i.e., nodes without a defined cause. Edges are then established by assessing the plausibility of connections between nodes. This staged approach incorporates new context into the graph, promoting scenario diversity, while enforcing logical correctness through common-sense reasoning. At this stage, the graph contains only event nodes, forming a causal graph with a directed acyclic structure. Users can specify a stopping criterion, such as a maximum number of nodes or graph depth; otherwise, the process continues indefinitely. Subsequent steps involve adding entity and property nodes to incorporate fine-grained details into the graph through a similar graph expansion process. The complete graph expansion process is outlined in Algorithm 1.

## 2.2 GROUNDING TO SIMULATED ENVIRONMENT

Given a subgraph $g \subseteq \mathcal{G}$ that forms a unique sequence of event nodes connected by event-to-event edges, the goal is to generate a simulation that specifies: (i) the initial state of the environment $q_0$, defined by the preconditions $P_{\text{init}}$, (ii) the motions of all dynamic actors, governed by the transition function $\delta : Q \times \Sigma \rightarrow Q$ of the abstract states $Q$, and (iii) the terminal conditions, characterized by the post conditions $P_{\text{term}}$ that define the symbolic constraints for the final states, including success or failure criteria, represented by the accepting states $F \subseteq Q$. Inspired by other work (Nguyen et al., 2024), we convert $g$ into a FSM configured with the same model and hyperparameters as in the graph expansion step. The FSM, formally defined as a tuple $\text{FSM} = (Q, \Sigma, \delta, q_0, F)$, serves as a symbolic representation of the task plan and is used to (i) verify its feasibility prior to execution, and (ii) trigger state changes based on simulation context, such as opening a car door when the ego-vehicle is nearby.

The set of abstract states $Q$ are high-level code abstractions, which bridges abstract reasoning with low-level simulation states. These abstractions are implemented as executable code compatible with the simulation engine. For example, an abstract state such as *"yielding to ambulance"* provides an abstraction for checking whether the ambulance is nearby and whether the ego-vehicle is stationary: `check_car1_near_car2('ambulance', 'ego') && is_stationary('ego')`. For further examples see Appendix A.2.

## 3 EXPERIMENTS

In this section, we start in Section 3.2 with qualitative analysis that demonstrates examples and capabilities of the inverse design approach; in Section 3.3, we compare diversity of the generated simulation against existing benchmarks or simulations; in Section 3.4, we showcase **ReGen** can be used for effective corner case generation; in Section 3.5, we demonstrate that the complexity

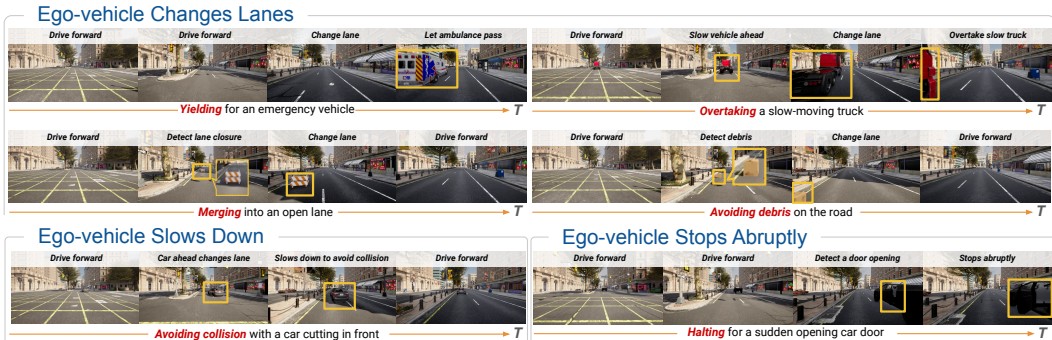

Figure 2: **ReGen for driving**. Given a behavior such as "changing lanes," our method can generate diverse simulated environments in which the behavior could have occurred, such as changing lanes to "*yield for an emergency vehicle*", "*overtaking a truck*", "*merging into an open lane*", or "*avoiding debris*." The bottom row illustrates additional cases: "*ego-vehicle slows down ← avoids collision*" and "*ego-vehicle stops ← halts for an opening car door.*"

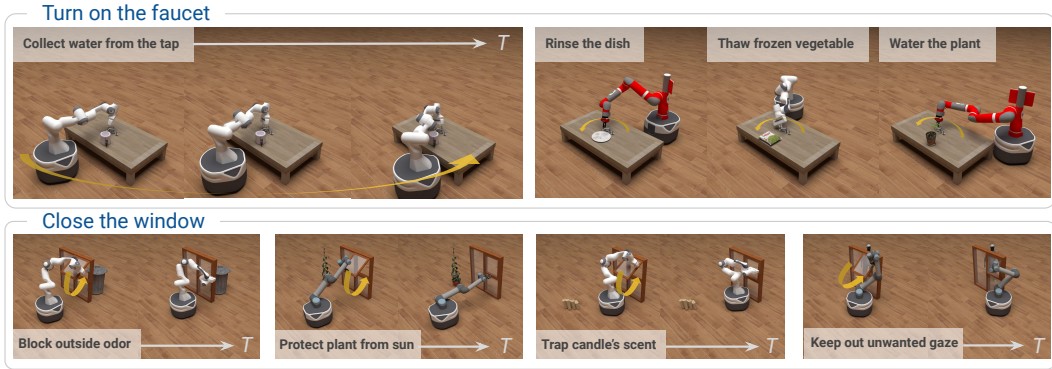

Figure 3: **ReGen for manipulation**. Given a robot manipulator behavior such as "turning on the faucet," our method generates diverse simulated environments with contextually appropriate scenarios, including "*collecting water,*" "*rinsing a dish,*" and "*watering a plant.*" In cases where certain physical effects—such as temperature changes in the "*thawing frozen vegetables*" scenario—are not explicitly modeled, our method simplifies the process by retrieving and appropriately placing objects in the environment.

of our generated environments allows us to create vision-language-action datasets that pose greater challenges to VLMs compared to existing datasets.

## 3.1 EXPERIMENT SETUP

**Driving.** For autonomous driving, we utilize the *CARLA* simulator (Dosovitskiy et al., 2017). We selected six key ego-motion behaviors: *driving forward, changing lanes, stopping at an intersection, stopping abruptly in the middle of the road, and stopping after making a right turn.* Each behavior is defined in natural language and mapped to a predefined route, encoded in an XML file that defines the start location, target waypoints, and speed. The ego-vehicle follows this route generated by an $A^*$ search algorithm, with longitudinal and lateral PID controllers for speed and steering control. To simulate other agents, we use motion primitives such as `stationary(location)` and `drivingforward(start, speed)`. These agents also use $A^*$ for path planning and PID controllers for low-level control. Parameter values (e.g., start locations, speeds) are determined using Google's CP-SAT solver (Perron & Furnon, 2019) to ensure compliance with the FSM's constraints.

**Manipulation.** For manipulation tasks, we use the PyBullet simulator (Coumans & Bai, 2016–2021) and selected 10 example reward functions from RoboGen (Wang et al., 2024d) for general tasks such as *closing the window* and *opening the door.* In PyBullet, behavior is limited to stationary interactions as the environment consists of non-articulated meshes that do not support dynamic motion. We use Google's the CP-SAT solver to determine parameter values (e.g. object location), and employ Soft Actor Critic (SAC) (Haarnoja et al., 2018) for policy training.

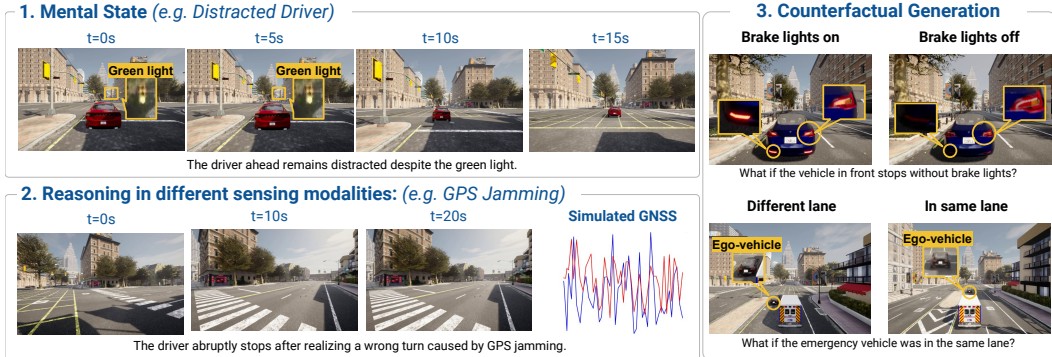

Figure 4: **Emergent capabilities of *ReGen*.** (1) Our method can simulate scenarios that capture the mental states or decision-making process of actors—for example, *a distracted driver failing to move despite a green light*. (2) Our method can model reasoning across different sensing modalities, such as simulating a scenario with *GPS jamming* with noisy GNSS measurements in CARLA. (3) Our method enables counterfactual scenario generation by perturbing the constructed graph with "what-if" questions—for example, changing from "*the front car stops with brake lights*" to "*the front car with broken brake light stops and thus brake light being off*."

## 3.2 QUALITATIVE ANALYSIS

**Generative Simulation via Inverse Design.** We demonstrate the inverse design approach of *ReGen* by generating diverse simulations from a single behavior, showcasing qualitative results across driving and manipulation domains. As shown for driving in Figure 2, when provided with a behavior such as "changing lanes," our method can simulate scenarios such as "*yielding for the ambulance*," "*overtaking a slow vehicle*," or "*avoiding debris on the road*." Unlike prior methods, which often required extensive data collection or expert-curated policies to create such scenarios, we demonstrate that our approach can generate these variations by simply reconfiguring the environmental context with respect to the given robot behavior. For instance, as shown in Figure 7 in the Appendix shows that ChatScene primarily generates collision avoidance scenarios, while DriveCoT and DriveLM focus mainly on general driving scenarios. In contrast, *ReGen* supports diverse task scenarios, including "*picking up passengers*," "*stopping for pedestrians*," and "*yielding to emergency vehicles*." This capability can potentially be applied to augment the simulation of the original robot learning pipeline, further enhancing the robustness of the learned behaviors.

Similarly, in the manipulation domain as in Figure 3, given an input such as "turn on the faucet," our method can infer new action verbs and their corresponding purposes, such as "*thawing* frozen vegetables," "*watering* a plant," or "*washing* dishes." This capability extends beyond simulating actions to also capturing the underlying intent or goal behind each action.

**Simulating Mental States.** Accurately capturing subtle mental states and decision-making processes from real-world driving datasets is inherently challenging. *ReGen* provides a framework for simulating nuanced mental states, such as a distracted driver at an intersection (see Figure 4 (1)). Previous work, like DriveCoT (Wang et al., 2024c), employs rule-based expert policies to control vehicles and generate ground truth labels for reasoning processes. However, this approach can introduce extraneous variables that obscure causal relationships. For example, in a DriveCoT scenario labeled as yielding at an intersection, $16/17$ annotations correctly attributed the stop to a traffic sign, but missed the emergency vehicle's influence as a contributing factor. *ReGen* mitigates this by reusing scenes and applying targeted interventions to simulate distinct cognitive processes, providing greater control in modeling mental states.

**Reasoning with Different Sensing Modalities.** *ReGen* extends its capabilities by reasoning over multiple data modalities, including vision, language, and other distinct sensor inputs such as GNSS (see Figure 4 (2)). By leveraging large language models (LLMs), *ReGen* can provide reasoning over these different modalities to simulate complex scenarios that influence the decision-making of the ego-driver. For instance, invoking functions such as add_gnss_noise() allow for the simulation of GPS jamming, connecting sensor noise to abstract concepts such as signal interference.

| Method | Number of Scenarios | Embedding Diversity ↑ | SelfBleu Diversity ↑ |
|---|---|---|---|
| NHTSA Crash Report | 24 | 0.1381 | 0.4350 |
| Zero-shot (gpt-3.5-turbo-0125) | 30 | $0.1509 \pm 0.01$ | $0.0945 \pm 0.03$ |
| Zero-shot (gpt-4o-2024-08-06) | 30 | $0.1680 \pm 0.02$ | $0.6082 \pm 0.13$ |
| Zero-shot (top-p=1, temp=1) | 30 | $0.1767 \pm 0.04$ | $\mathbf{0.7814 \pm 0.06}$ |
| Zero-shot (top-p=0, temp=0) | 30 | $0.1493 \pm 0.01$ | $0.4143 \pm 0.02$ |
| ChatScene | 40 | 0.1214 | 0.2945 |
| DriveLM | 696 | $0.1135 \pm 0.00$ | $0.4731 \pm 0.01$ |
| ***ReGen*** (Ours) | 24 | **0.2268** | 0.7377 |

Table 1: **Simulation diversity for driving.** The baselines include NHTSA typology (National Highway Traffic Safety Administration, 2007), a zero-shot LLM method, ChatScene (Zhang et al., 2024) (few-shot), and DriveLM (Sima et al., 2024). Apart from expert driving or exclusively safety-critical scenarios, our method can generate a broader range of environments, e.g., yielding to emergency vehicles, navigating intersections with malfunctioning traffic light, thus achieving better diversity.

| Methods | # Environments | Unique Reward Functions | Embedding Diversity ↑ |
|---|---|---|---|
| Behavior-100 | 100 | 100 | $0.5513 \pm 0.01$ |
| RLBench | 106 | 106 | $0.5819 \pm 0.01$ |
| GenSim | 152 | 152 | $0.4350 \pm 0.01$ |
| RoboGen (manipulation) | 46 | 46 | $0.5787 \pm 0.01$ |
| RoboGen (subset) | 10 | 10 | 0.5536 |
| ***ReGen*** (Ours) | 38 | 10 | **0.6560** |

Table 2: **Simulation diversity for manipulation.** We compare to Behavior-100 (Srivastava et al., 2021), RLBench (James et al., 2019), GenSim (Wang et al., 2024b), and RoboGen (Wang et al., 2024d). Our method augment simulation in an orthogonal axis that changes environmental context, e.g., "opening the door to let the pet in" or "opening the door to pick up delivery", as opposed to purely skill-driven environments, e.g., "opening the door", thus achieving better diversity.

**Counterfactual Generation.** Our method can generate counterfactuals, such as modifying brake lights or adjusting the initial location of surrounding vehicles, to improve explainability in multimodal foundation models (Figure 4 (3)). For example, by altering brake lights, we can test whether the model infers speed from visual motion cues or relies on brake lights as a shortcut, addressing the limitation of multimodal foundation models identified in prior work (Sreeram et al., 2024).

### 3.3 Simulation Diversity

In this section we conduct a series of experiments to evaluate the diversity of generated simulations against extensive baselines in Table 1 and 2. To quantify the diversity in terms of task semantics and scene configurations, we use text diversity metrics, following approaches from (Wang et al., 2024d; Nguyen et al., 2024). Specifically, we assess diversity using metrics such as Self-BLEU (Zhu et al., 2018; Papineni et al., 2002) and embedding similarity with Sentence-BERT (Reimers & Gurevych, 2019). For each method, we sample a set equal to the smallest sample size among the baselines and compute their similarity score, repeating this process 10 times and reporting their average diversity score as $1 - \text{similarity}$.

**Driving.** Table 1 shows the scenario diversity results in the driving domain. We compare our method against CARLA Leaderboard 2.0. scenarios, which cover traffic scenarios based on the NHTSA typology (National Highway Traffic Safety Administration, 2007). ChatScene (Zhang et al., 2024) employs few-shot prompting of large language models to generate diverse safety-critical scenarios, while DriveLM (Sima et al., 2024) provides graph annotations from the NuScenes dataset (Caesar et al., 2020). Additionally, we evaluate the diversity of zero-shot methods across varying top-p and temperature settings, as well as with different LLM models, including gpt-4o-2024-08-06 and gpt-3.5-turbo-0125. Our method consistently outperforms all baselines in scenario diversity, as measured by embedding similarity and Self-BLEU scores. Methods like ChatScene focus exclusively on safety-critical scenarios, while DriveLM is limited to general driving scenarios. Although increasing the top-p and temperature settings slightly improves diversity for the zero-shot baseline, our method achieves superior diversity even with conservative settings (top-p = 0 and temperature = 0), demonstrating that the improvements are not merely a result of tuning these pa-

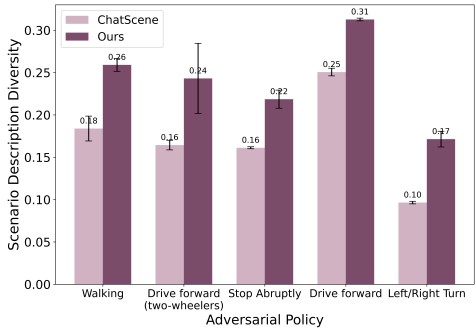

Figure 5: **Diversity of Corner Cases.** Compared to ChatScene (Zhang et al., 2024), our method generates more diverse corner cases via reasoning about different causes.

| Metric | Method | Base Traffic Scenarios | | | |
|---|---|---|---|---|---|
| | | SO | TO | LaneC | VP |
| CR ↑ | LC | 0.30 | 0.09 | 0.87 | 0.83 |
| | AdvSim | 0.51 | 0.33 | 0.86 | **0.87** |
| | CS | 0.45 | 0.61 | 0.89 | **0.87** |
| | AdvTraj | 0.50 | 0.31 | 0.78 | 0.82 |
| | ChatScene | 0.89 | 0.70 | 0.95 | 0.79 |
| | ReGen (ours) | **0.90** | **0.83** | **0.96** | 0.77 |

Table 3: **Collision Rate in SafeBench.** SO means Straight Obstacle. TO means Turning Obstacle. LaneC means Lane Changing. VP means Vehicle Passing. Baselines include LC (Ding et al., 2020), AdvSim (Wang et al., 2023b), CS (Wang et al., 2023b), AdvTraj(Cao et al., 2022), and ChatScene (Zhang et al., 2024). Our method can produce corner cases that lead to a higher collision rate (CR).

rameters. In contrast to the baselines, our approach can simulate a broader range of scenarios, such as yielding to emergency vehicles or navigating intersections with malfunctioning traffic lights. The success rate of our scenario generation is $80\%$, with a detailed breakdown provided in Table 6 in the Appendix. Most failure cases are due to overly strict FSM constraints that require multiple conditions to be satisfied simultaneously. Although semantically correct, it imposes unnecessary strict satisfiability requirements. Further discussion can be found in Appendix A.4.2.

**Manipulation.** Table 2 shows the simulation diversity in the manipulation domain. Since the task descriptions were short we only report the embedding diversity since $n$-gram based metrics are not applicable. We compare our results against baselines such as RoboGen (Wang et al., 2024d), GenSim (Wang et al., 2024b), Behavior-100 (Srivastava et al., 2021), and RLBench (James et al., 2019). In prior work, generating a manipulation simulation often required learning a new task either through reward design (Wang et al., 2024d) or by using expert demonstrations (Wang et al., 2024b). We demonstrate using our inverse design framework that we can generate simulations by reusing learned behaviors from these methods to simulate new scenarios. We use 10 reward functions from RoboGen, each generating 5 variants of environments. With success rate of $78\%$, we end up having 38 environments. The most failure cases are invalid reasoning of articulated objects, e.g., a shelf-opening event with the asset not being able to open. Among all baselines, our method achieve the highest diversity. This is mainly because prior works mostly focus on simulating environment corresponding to a single skill, often within similar contexts – e.g., "open the door" and "close the door", while our method augments simulation in an orthogonal axis such as "open the door to let the pet in." However, unlike driving where fine-grained control on many entities especially dynamic actors is possible, our approach simplifies manipulation tasks by limiting actions to retrieving and placing entities in contextually appropriate locations (e.g., the dog has to be outside the door). Elements not supported by the simulation engine – such as simulating the motion of a pet walking in – are skipped. We leave these (better articulated objects and more flexibility of simulation engine for manipulation) as future research.

## 3.4 CORNER CASE GENERATION

In Figure 5, we demonstrate our method's ability to generate diverse corner case scenarios for testing in safety critical scenarios, as detailed in Table 3. We highlight *ReGen*'s ability to reuse an existing adversarial policy behavior in more diverse context. For comparison, we evaluated against ChatScene (Zhang et al., 2024), which uses LLM few-shot prompting and converts text into simulation via text-to-scenic. For fair comparison, we use the same adversarial policy behavior as theirs but change the context of the environment. We consistently achieved greater diversity than ChatScene across all adversarial policy behaviors. While ChatScene introduces only minor variations, such as a pedestrian crossing in front of a car or vending machine, our method captures a broader range of cause-and-effect relationships. For the "pedestrian walking" behaviors, our method generates unique causes, such as a group of protesters causing a collision or a single pedestrian forcing another vehicle to stop abruptly in front of the ego-vehicle. These scenarios underscore the greater diversity enabled by our approach. We observe that both methods achieve greatest diversity for the behavior "drive forward," as it can be applied in a broader range of contexts, such as driving fast or

| Method | Accuracy (%) ↓ | | |
|---|---|---|---|
| | GPT-4o | GPT-4-Turbo | Claude 3.5 Sonnet |
| DriveCoT | 0.87 | 0.80 | 0.80 |
| DriveLM | 0.90 | 0.83 | 0.77 |
| ReGen (Ours) | **0.63** | **0.53** | **0.50** |

Table 4: **Complex Vision-language-action Dataset from Our Simulated Environments.** We compare with existing driving-related reasoning datasets, DriveCoT (Wang et al., 2024c) and DriveLM (Sima et al., 2024). The accuracy of reasoning about the correct ego-vehicle actions is reported. Our dataset consists of more complex scenarios compared to baselines with mostly urban driving maneuvers, thus posing greater challenges to existing VLMs.

slow, accelerating, and decelerating. In contrast, both methods exhibit lower diversity for left and right turns, as these scenarios offer fewer plausible contextual variations.

We then evaluated these generated scenarios against ChatScene and other adversarial policy learning baselines including Learn-to-Collide (LC) (Ding et al., 2020), AdvSim (Wang et al., 2023b), Carla Scenario Generator (CS) (Wang et al., 2023b), Adversarial Trajectory Optimization (Cao et al., 2022) to test whether the corner-case simulations led to higher collision rates. The results, presented in, Table 3, only includes the collision rates, as the final score incorporates driving infractions, which fall outside the scope of generating collision scenarios. To run the benchmark, we converted our scenarios into scenic files, required in the SafeBench benchmark (Xu et al., 2022). This process was done manually as generating scenic files is outside the scope of our work.

Both our method and ChatScene outperform other baselines by leveraging LLMs to intelligently spawn adversarial agents. We further validate by human inspection that all generated environments do not place actors in the locations that cause inevitable collision, namely too close to the ego-car. However, our method surpasses ChatScene by diversifying the underlying causes of challenging scenarios. While ChatScene primarily generates scenarios where an object crosses in front of the ego-vehicle (e.g., a pedestrian crossing or a car merging), our method introduces more complex variations, such as a group of pedestrians or a vehicle traveling in the opposite direction. Empirically, these scenarios are significantly more challenging for the driving policies, such as requiring larger steering adjustments to avoid groups of pedestrians.

## 3.5 PROBING MULTIMODAL FOUNDATION MODELS

In this section, we evaluate how well the state-of-the-art vision language models (VLMs) can reason about the vision-language-action dataset produced by the generated simulation from our methods, compared to existing datasets. We aim to (i) demonstrate a scalable possibility of multi-modal data synthesis with complex reasoning and (ii) use the complexity of the produced dataset as an indirect measure to demonstrate the complexity of the generated simulation. We conduct experiment in the driving domain; the goal of the VLMs is to process a sequence of images along with textual context and question, and answer most plausible actions to be taken by the ego-vehicle. Specifically, we assess the VLMs' ability to infer the desired action in our generated driving scenarios and compare their performance with two literature baselines: DriveCoT (Wang et al., 2024c) and DriveLM (Sima et al., 2024). In DriveCoT the authors used rule-based expert policies to control ego and generated ground-truth labels for reasoning processes, while in DriveLM they employed graph annotations on NuScenes dataset (Caesar et al., 2020) as a large scale real-world driving dataset. From each dataset, we randomly select 30 simulation traces, together with the ground-truth desired ego action for each one. As part of the preprocessing, we extract three consecutive key frames from each simulation trace, where the last key frame corresponds to the timepoint where the desired ego action has been recorded. Since here we are solely evaluating the planning capability of VLMs, we also provide some privileged information, such as the location and speed of all entities in the scene, to the VLMs. Therefore, the three key frames, along with their corresponding privileged information that are parsed from the recorded log files, are provided to the VLMs, and they are prompted to identify the desired action for ego. The VLMs tested in this evaluation include GPT-4o (OpenAI, 2024), GPT-4-turbo (Achiam et al., 2023), and Claude 3.5 Sonnet (Anthropic, 2024). The success rates of the VLMs in inferring the desired actions are presented in Table 4. As shown in Table 4, the success rate of VLMs in inferring the desired ego actions for our dataset is significantly lower compared to the DriveCoT and DriveLM datasets. This difference is due to the greater diversity of

scenarios generated by our method compared to those in DriveCoT and DriveLM. In the baseline datasets, the desired ego actions are primarily common urban driving maneuvers, such as stopping or moving forward, whereas our method produces scenarios with a wider range of ego actions.

We observed that VLMs frequently responded with deceleration as the default action, consistent with findings in prior work (Sreeram et al., 2024). For instance, in lane change scenarios such as "avoiding debris", "overtaking a slow vehicle", "merging into an open lane", or "swerving to avoid a wrong-way driver" the VLM often suggested that the ego-vehicle should brake and stop behind obstacles instead of performing a logical lane change to avoid them. These outcomes suggest that VLMs generally struggle in environments with nuanced spatial and situational reasoning. In contrast, DriveCoT (also using the CARLA simulator) generates scenarios where objects appear directly in the ego-vehicle's path, requiring a deceleration response. In such cases, the VLMs' biases align with the expected behaviors for those scenarios. These observed failures underscore a limitation of off-the-shelf VLMs in reasoning about complex driving scenarios. Conversely, the scenarios in DriveLM were less challenging for VLMs, as they mostly comprised general driving scenarios, as shown in Appendix A.4.1.

## 4 RELATED WORK

**Robot Simulation.** Simulation has played a critical role for general robotics. Simulated environments for driving (car, 2020; Amini et al., 2022; Gulino et al., 2024), manipulation (Zhu et al., 2020; James et al., 2020; Nasiriany et al., 2024a), and locomotion (Rudin et al., 2022; Makoviychuk et al., 2021; Wang et al., 2023d), have each contributed significantly to their robotic subfield. Their use for verification has its own community and field of research (Kleijnen, 1995; Pace, 2004; Corso et al., 2021), in parallel to significant efforts for leveraging such approaches for policy training (Muratore et al., 2022; Wang et al., 2022; Loquercio et al., 2019). Our work focuses on a new paradigm of robot simulation that uses generative models for more scalable simulation construction.

**Generative Simulation.** Generative simulation methods have emerged as a powerful tool for automatically creating diverse and realistic scenarios in robot manipulation (Wang et al., 2023c; Mandlekar et al., 2023) and more general robotic tasks (Wang et al., 2024d; Nasiriany et al., 2024b). Our work shares the same goal of automating the process of constructing simulation with the power of generative AI. Uniquely, we follow an inverse design approach which is more tailored for validation and augmentation use cases, complementing existing approaches with an orthogonal axis.

**LLMs for simulation.** Specific approaches for diverse data generation (Shiroshita et al., 2020; Sinha et al., 2020) within simulation environments, including those that leverage language models (Zhong et al., 2023; Elmaaroufi et al., 2024; Aasi et al., 2024) are playing an important role in how simulation environments are used for both verification and system training. More broadly, our work relates to the rich set of different approaches harnessing LLMs for reasoning about, and planning in, domains such as robotics (Zeng et al., 2023; Wang et al., 2024a) and autonomous driving (Cui et al., 2023). These include aspects such as language-based planners (Song et al., 2023; Liu et al., 2023; Mao et al., 2023), and simulation environments (Zala et al., 2024), among others.

## 5 CONCLUSION

We present an inverse design approach for generative simulation, demonstrated in autonomous driving and manipulation. Using LLMs and simulation engines, we construct and expand graphs that capture cause-and-effect relationships and relevant entities, which are then converted into simulated environments. With the extensive experiments, we demonstrate capabilities of our method via qualitative analysis, superior diversity compared to existing simulation, more effective corner-case generation, and more complex vision-language-action dataset synthesis than current dataset.

**Limitation.** There are properties that our method can reason about but are not fully simulatable due to limitations of the simulation engine. For example, without simulating temperature, we cannot measure the progress of "thawing the frozen vegetable". This limitation is more common in manipulation tasks than in driving, where robots primarily interact with other actors, allowing for more flexible simulation. Another challenge is handling articulated objects, such as reasoning about opening a shelf, which cannot be simulated with the available assets; more details in Appendix A.4.2.

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

# A  APPENDIX

## A.1  GRAPH EXPANSION DETAILS

### A.1.1  ASSET DATABASE

The simulator database, $\mathcal{D}_{\text{asset}} = (V, E)$, is represented as a directed graph, where each node $v \in V$ corresponds to a simulatable asset, such as sensors (e.g. GPS, temperature, humidity), agents (entities with a dynamic model), objects (static entities without a dynamic model), properties (e.g., siren, car door, light color), and behaviors of other agents (e.g., driving forward, standing still, walking). Each edge $e \in E$ represents a relationship between two assets, such as (siren → ambulance), indicating that the siren is a property of the ambulance. A property is a node that has at least one outgoing edge, while agents and objects only have incoming edges. We provide an example of $\mathcal{D}_{\text{asset}}$ for the CARLA simulator and PyBullet:

**Code Example 1: Asset Database for CARLA Simulator ($\mathcal{D}_{\text{asset}}$)**

```
node["vehicles"] = ["bicycle", "sedan", "ambulance"]
node["traffic"] = ["traffic light"]
node["behavior"] = ["constant speed/stationary/change Lanes..."]
node["traffic light"] = ["red/green/yellow/off"]
node["ambulance"] = ["siren", "behavior"]
node["bicycle"] = ["behavior"]
node["sedan"] = ["behavior", "front door"]
node["siren"] = ["on/off"]
node["front door"] = ["open/closed"]
node["constant speed"] = ["ending location", "starting location",
"target speed"]
...
```

Here, an entity such as an *ambulance*, has properties such as ['siren' , 'behavior']. In CARLA, th siren property can have states—on or off—indicated by /. Meanwhile, the *behavior* property, such as *constant speed*, includes an *ending location* property. This location, for example, '*in front of the ego-vehicle*,' can be queried from the LLM.

**Code Example 2: Asset Database for PyBullet ($\mathcal{D}_{\text{asset}}$)**

```
node["static objects"] = ["desk lamp", "tv", "trash can", "ceramic cup", "book",
    "toy", "children", "adult", ...]
node["behavior"] = ["stationary/stationary"]
node["stationary"] = ["location"]
...
```

In the PyBullet simulator, the behavior is limited to *stationary* because the environment consists soley of non-articulated meshes, which do not support dynamic motion.

### A.1.2  NODE PROPOSAL

**Event nodes.** Candidate nodes can be generated either by an LLM or provided by the user. For instance, in Example 1, given a source node, the LLM generates plausible events, such as 'stopping because there is a jaywalker.' In another example, if the prior involves a 'police car,' the generated events might include scenarios like a 'police chase.' However, not all nodes proposed during the node proposal stage are valid causes. For instance, 'animal on the road' might be invalid if it is not simulatable, or 'a jaywalker in another city' would be logically implausible. During edge construction, each proposed node is validated to ensure it represents a plausible cause of the source node.

---

**Example 1: Event nodes ($\mathcal{V}_{\text{event}}$)**

```
Source node: "The ego-vehicle stopped abruptly"
***Example #1***
Prior: null
Candidate event nodes (proposed by LLM): ["a jaywalker walked in front", "animal
    on the road", "emergency vehicle approaching from behind", "debris ahead"]

***Example #2***
Prior: "police car"
Candidate event nodes (prosposed by LLM): ["road block", "police chase",
    "arrest", ...]

***Example #3***
Candidate event nodes (proposed by user): ["a tree fell in front", "a jaywalker
    in another city", "a cyclist changed lanes"]
```

---

**Entity nodes** From the asset database $\mathcal{D}_{\text{asset}}$, the candidate nodes include all entities listed in $\mathcal{D}_{\text{asset}}$, starting from the node node[``vehicles"]. In the implementation, we also append additional entities, such as pedestrians and static objects. The appropriate vehicle for the source node will be selected during the edge proposal stage, detailed in Appendix A.1.3.

---

**Example 2: Entity nodes ($\mathcal{V}_{\text{entity}}$)**

```
Source node: "Emergency vehicle approaching from behind"
Candidate event nodes: ["bicycle", "ambulance", "sedan"]
```

---

**Property nodes** In Example 1, we verify whether the variable exists in the databaes. For instance, the variable 'siren' is found in $\mathcal{D}_{\text{asset}}$, allowing us to retrieve its values such as 'on' and 'off,' as detailed in Appendix A.1.1. In Example 2, the variable 'location' does not have predefined values (i.e. no 'location' key in node) so we query the LLM to generate all possible locations. However, not all generated locations may be valid. The appropriate nodes are selected during the edge proposal stage.

---

**Example 3: Property nodes ($\mathcal{V}_{\text{property}}$)**

```
Event-to-event graph: "ego-vehicle stopped abruptly <- emergency vehicle
    approaching from behind"
Entity-to-event graph: "emergency vehicle approaching from behind <- ambulance"

***Example 1***
Source node: "siren"
Candidate event nodes (from asset database): ["on", "off"]

***Example 2***
Source node: "start location"
Candidate event nodes (proposed by LLM): ["behind the ego-vehicle on adjacent
    lane", "behind the ego-vehicle on same lane", "in front of ego-vehicle on
    adjacent lane", "in front of ego-vehicle on same lane"]
```

---

### A.1.3  EDGE CONSTRUCTION

Given a list of candidate nodes (from Appendix A.1.2), the edge construction process is used to select plausible nodes and eliminate unlikely ones. For event-to-event edges, a node such as "a jaywalker in another city" can be excluded because the causal relationship "ego-vehicle stopping abruptly ← a jaywalker in another city" is implausible. For entity-to-event edges, only nodes such as "ambulance" are selected from the available entities, as they are relevant to the source node "emergency vehicle approaching from behind." In cases where the source node is "tree falls in front," and no corresponding "tree" entity exists in the simulator, no edges are created. Finally, for property-to-entity edges, relevant simulatable properties are selected, such as "siren" for an ambulance.

**Example 4: Event-to-event ($\mathcal{E}_{event}$)**

```
Source node: "ego-vehicle stopped abruptly"
Candidate nodes: ["a jaywalker walked in front", "animal on the road",
    "emergency vehicle approaching from behind", "debris in the road", "a tree
    fell in front", "a jaywalker in another city", "a cyclist changed lanes"]

**Chosen nodes**: ["a jaywalker walked in front", "animal on the road",
    "emergency vehicle approaching from behind", "debris in the road", "a tree
    fell in front", "a cyclist changed lanes"]
**Nodes not chosen**: ["a jaywalker in another city"]
**Generated graphs**:
    1. ego-vehicle stopped abruptly <- a jaywalker walked in front
    2. ego-vehicle stopped abruptly <- animal on the road
    3. ego-vehicle stopped abruptly <- emergency vehicle approaching from behind
    4. ego-vehicle stopped abruptly <- debris in the road
    5. ego-vehicle stopped abruptly <- a tree fell in front
    6. ego-vehicle stopped abruptly <- cyclist changed lanes
```

**Example 5: Entity-to-event ($\mathcal{E}_{entity}$)**

```
***Example 1***
Source node: "Emergency vehicle approaching from behind"
Candidate event nodes: ["bicycle", "ambulance", "sedan"]

*Chosen nodes*: ["ambulance"]
*Nodes not chosen*: ["bicycle", "sedan"]
*Generated graphs*:
    1. emergency vehicle approaching from behind <- ambulance

***Example 2***
Source node: "Tree fell in front"
*Chosen nodes*: []
*Nodes not chosen*: ["bicycle", "ambulance", "sedan"]
Return that this event cannot be simulated in CARLA
```

**Example 6: Property-to-entity ($\mathcal{E}_{property}$)**

```
Event-to-event graph: "ego-vehicle stopped abruptly <- emergency vehicle
    approaching from behind"
Entity-to-event graph: "emergency vehicle approaching from behind <- ambulance"

***Example 1***
Source node: "siren"
Candidate event nodes (from asset database): ["on", "off"]

*Chosen nodes*: ["on"]
*Nodes not chosen*: ["off"]
*Generated graphs*:
    1. ambulance <- siren

***Example 2***
Source node: "start location"
Candidate event nodes (proposed by LLM): ["behind the ego-vehicle on adjacent
    lane", "behind the ego-vehicle on same lane", "in front of ego-vehicle on
    adjacent lane", "in front of ego-vehicle on same lane"]

*Chosen nodes*: ["behind the ego-vehicle on adjacent lane"]
*Nodes not chosen*: ["behind the ego-vehicle on same lane", "in front of
    ego-vehicle on adjacent lane", "in front of ego-vehicle on same lane"]
*Generated graphs*:
    1. behind the ego-vehicle on same lane <- starting location <-- ambulance ...
```

The graph expansion process produces a graph that defines an environment and describes the scenario. This graph serves as input to the LLM to generate the Low-Level State Translator (LLST), which bridges abstract reasoning with physical state transitions in order to track states. This tracking is crucial for defining constraints that align with the intended scenario we aim to simulate. For example, the abstract state "Ambulance Approaching" defines a constraint that requires the ambulance to be behind the ego-vehicle and in motion.

**Code Example 3: Generated Graph**

```
causal_graph = ['Ambulance approaching from behind', 'Ego-vehicle abruptly
    stopped on left lane']

entities = [{'name': 'ambulance1', 'type': 'agent', 'entity_name': 'ambulance',
    'behavioral_properties': {'action': 'Vehicle drives straight the entire
    time', 'starting location': 'behind the ego vehicle in the right lane',
    'ending location': 'in front of ego vehicle in the right lane'}},
{'name': 'ego-vehicle', 'type': 'agent', 'entity_name': 'ego-vehicle',
    'behavioral_properties': {'action': 'Vehicle drives straight and suddenly
    stops'}}]
```

**Code Example 4: Low-Level State Translator**

```
def _agent_state_tracker(self, agent_name) -> None:
    if agent_name == "ambulance1":
        # State: Ambulance Approaching
        if behind_vehicle(agent_name, "ego-vehicle") and
    is_currently_moving(agent_name):
            self._update_state("Ambulance Approaching", agent_name, True)

        # State: Ambulance Close to Ego
        if are_close_by(agent_name, "ego-vehicle") and
    is_currently_moving(agent_name):
            self._update_state("Ambulance Close to Ego", agent_name, True)

    ...
```

These abstract states are subsequently used to construct a finite state machine (FSM), incorporating transitions that capture the temporal dynamics of the scenario and encode temporal logic. For example, in this scenario, the abstract state "Ambulance Approaching" must occur and must precede the state "Ambulance Passing Ego."

**Code Example 5: Finite State Machine**

```
    fsm = [[('ambulance1', 'Ambulance Approaching'), ('ego-vehicle', 'Ego
    Driving Steady')],
        [('ambulance1', 'Ambulance Close to Ego')],
        [('ego-vehicle', 'Ego Braking')],
        [('ego-vehicle', 'Ego Stopped Abruptly')],
        [('ambulance1', 'Ambulance Passing Ego')]]
```

Given a FSM, we use Google's CP-SAT solver to find solutions for the variables such as the $x$, $y$ coordinates of the start and end positions $(x_0, y_0, x_T, y_T)$, as well as the speed, such that it satisfies the constraints imposed by the FSM. For instance, the behavior of the ambulance, defined as "drive straight," is generated as: DriveStraight('ambulance', $x_0$, $y_0, x_T, y_T$, speed). The simulation considered valid only if it terminates in a state that satisfies the terminal condition of the FSM.

```
start": {"x": -25, "y": 4}, "end": {"x": 80, "y": 4}, "speed": 40,
```

**Full Config Example.** We provide the full example of the generated scenario config file below:

**Code Example 6: Scenario Config Example**

```
narrative = "An ambulance approached from behind, prompting the ego vehicle to
    stop abruptly, allowing the ambulance to pass safely."

entities = [{'name': 'ambulance1', 'type': 'agent', 'entity_name': 'ambulance',
    'behavioral_properties': {'action': 'Vehicle drives straight the entire
    time', 'starting location': 'behind the ego vehicle in the right lane',
    'ending location': 'in front of ego vehicle in the right lane'}}, {'name':
    'ego-vehicle', 'type': 'agent', 'entity_name': 'ego-vehicle', 'properties':
    {}, 'behavioral_properties': {'action': 'Vehicle drives straight and
    suddenly stops'}}]

vehicles = [{"name": "ambulance1",
        "start": {"x": -25, "y": 4}, "end": {"x": 80, "y": 4},
        "speed_range": [40, 40],
        "blueprint_id": "vehicle.ford.ambulance",
        "driving_policy": "drive forward",
        "type": "dynamic",
        "heading": 0}]

causal_graph = ['Ambulance approaching from behind', 'Ego-vehicle abruptly
    stopped on left lane']

fsm = [[('ambulance1', 'Ambulance Approaching'), ('ego-vehicle', 'Ego Driving
    Steady')],
        [('ambulance1', 'Ambulance Close to Ego')],
        [('ego-vehicle', 'Ego Braking')],
        [('ego-vehicle', 'Ego Stopped Abruptly')],
        [('ambulance1', 'Ambulance Passing Ego')]]

class StateManager(StateManagerBase):
    def __init__(self, obj_name, world):
        super().__init__(obj_name, get_object_states(), world)

    def _agent_state_tracker(self, agent_name) -> None:
        if agent_name == "ambulance1":
            # State: Ambulance Approaching
            if behind_vehicle(agent_name, "ego-vehicle") and
    is_currently_moving(agent_name):
                self._update_state("Ambulance Approaching", agent_name, True)

            # State: Ambulance Close to Ego
            if are_close_by(agent_name, "ego-vehicle") and
    is_currently_moving(agent_name):
                self._update_state("Ambulance Close to Ego", agent_name, True)

            # State: Ambulance Passing Ego
            if right_in_front(agent_name, "ego-vehicle") and
    is_currently_moving(agent_name):
                self._update_state("Ambulance Passing Ego", agent_name, True)

        elif agent_name == "ego-vehicle":
            # State: Ego Driving Steady
            if is_ego_driving_steady(agent_name):
                self._update_state("Ego Driving Steady", agent_name, True)

            # State: Ego Braking
            if is_braking(agent_name):
                self._update_state("Ego Braking", agent_name, True)

            # State: Ego Stopped Abruptly
            if is_currently_stopped(agent_name):
                self._update_state("Ego Stopped Abruptly", agent_name, True)
        return None
```

## A.3   ABLATION

**Accuracy of edge creation.** In Table 5, we present an ablation study evaluating the accuracy of edge creation during the graph expansion stage. For event-to-event edges, we assess whether the LLM can correctly identify causal and non-causal variables given the input behavior, testing only at a depth of 1. For entity-to-event edges, we evaluate whether the LLM can accurately identify simulatable events and select corresponding assets from the database. The results shown in Table 5 consistently demonstrate high performance for event-to-event and entity-to-event edges across both domains, with accuracy exceeding 0.90. The consistency in LLM responses across trials can be attributed to the temperature and `top-p` hyperparameters being set to 0. Failure cases for entity-to-event stem from the LLM's misunderstanding of the simulator's capabilities, as described in $\mathcal{D}_{\text{asset}}$. For example, given a graph such as "ego-vehicle stopping ← flood to knees," the LLM might propose changing the weather (treated as an entity) to "flooding." While this is valid in general, it misaligns with the constraints of the simulation engine. Finally, for property-to-entity edges, we evaluate the LLM's ability to select the most plausible location. While the LLM performs well on simpler properties, such as determining whether a siren should be on, its accuracy decreases for properties requiring complex spatiotemporal reasoning. In our driving experiment, the LLM was tasked with selecting end locations for two entities across two scenarios, with each entity having 8 possible locations. We observed significantly higher variance in its responses.

|  |  | **Accuracy** | **Precision** | **Recall** | **F1 Score** |
|---|---|---|---|---|---|
| Event-to-Event | Driving | $0.98 \pm 0.04$ | $1.00 \pm 0.00$ | $0.97 \pm 0.06$ | $0.98 \pm 0.03$ |
|  | Manipulation | $0.98 \pm 0.04$ | $1.00 \pm 0.00$ | $0.97 \pm 0.04$ | $0.98 \pm 0.02$ |
| Entity-to-Event | Driving | $0.91 \pm 0.02$ | $0.86 \pm 0.02$ | $0.98 \pm 0.03$ | $0.92 \pm 0.01$ |
|  | Manipulation | $0.94 \pm 0.02$ | $1.00 \pm 0.00$ | $0.89 \pm 0.03$ | $0.94 \pm 0.02$ |
| Property-to-Event | Driving | $0.81 \pm 0.09$ | $0.73 \pm 0.11$ | $0.90 \pm 0.03$ | $0.80 \pm 0.08$ |
|  | Manipulation | $1.00 \pm 0.00$ | $1.00 \pm 0.00$ | $1.00 \pm 0.00$ | $1.00 \pm 0.00$ |

Table 5: **Accuracy of Edge Creation**

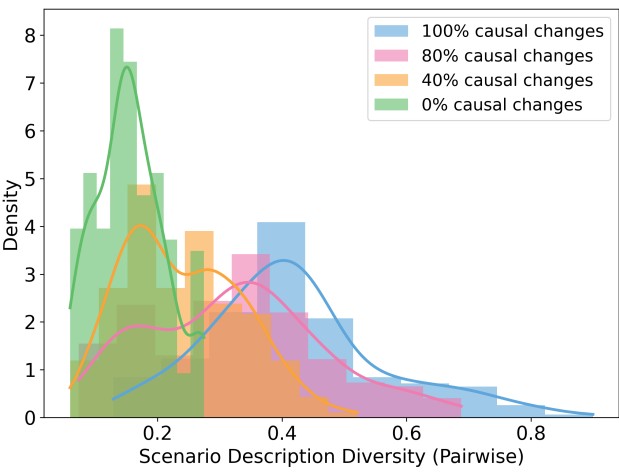

Figure 6: **Controllability**

**Eliciting Diversity & Controllability.** Figure 6 shows the pairwise diversity distribution of generated scenarios, measured as the proportion of scenarios with unique event-to-event nodes, excluding the input behavior node. A value of $100\%$ indicates that all compared scenarios have distinct causes, while $0\%$ means they share the same cause but vary in properties, such as their start location or behavior. We observe that scenario diversity increases as the causes vary. Notably, the bimodal distribution suggests that introducing a new causal graph introduces greater diversity by exploring broader cause-and-effect relationships, while changes to the property graph result in more subtle variations. By explicitly introducing new causal variables, we guide the model to explore a more diverse range of plausible outcomes, uncovering interactions that are otherwise not immediately apparent, but are entirely plausible.

**Re-usability.** Beyond generating diverse simulations, our method allows for the efficient reuse of existing behaviors. In Table 2, we show that the diversity of manipulation tasks generated by our method surpasses all baselines, even with the use of only 10 reward functions—a subset of RoboGen tasks. Although RoboGen supports a broad range of tasks, our method further increases task diversity by $18.50\%$. Previous methods encountered a bottleneck in simulating actions from text due to the need to design a unique reward function for every task (Wang et al., 2024d; Ma et al., 2024; Nguyen et al., 2024). Our method addresses this by changing the context to reflect different higher-level goals, enabling the creation of new tasks while reusing the same task-specific reward function. For instance, the task "opening a door" can be augmented to convey new narratives such as "ventilating a room," "letting the guest out," or "letting the pet in." This contextual dimension is a unique capability of ours that addresses the limitations of simulating actions from text.

### A.4 CASE STUDY

#### A.4.1 GENERATED SCENARIOS

**Driving.** Our method provides broader coverage of driving scenario categories than existing approaches, as shown in Figure 7. While methods like ChatScene primarily focuses on safety-critical scenarios and DriveLM covers general driving tasks, our approach generates a diverse range of scenarios that encompass both safety-critical and general driving situations, such as "yielding to an emergency vehicle" or "picking up a passenger."

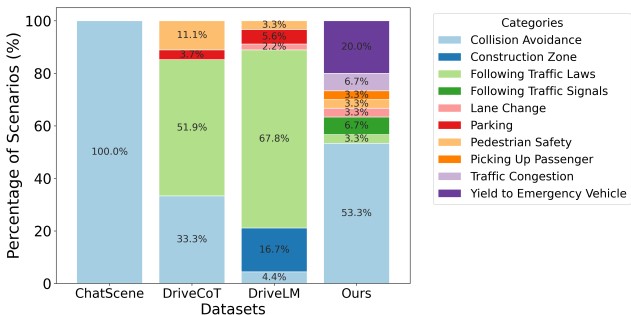

Figure 7: Categories of Scenarios (driving)

#### A.4.2 FAILURE MODES

Most failures in simulating the scenarios are from overly strict FSM constraints that required multiple conditions to be satisfied simultaneously. For example, in the scenario "*an object falling off a truck causing the ego-vehicle to stop*," we successfuly found solutions where the box falls to the ground and the ego-vehicle brakes. However, the scenario became infeasible because the FSM also required the delivery truck to exit the scene at the exact moment the ego-vehicle stopped. While this requirement is not inherently incorrect, relaxing the constraint—allowing the delivery truck's exit to occur after the ego-vehicle stopped—would make the scenario feasible.

> **Code Example 7: Failure case (driving)**
> ```
> fsm = [[('delivery_truck', 'Approaching Intersection'), ('box', 'On Truck'),
>     ('ego_vehicle', 'Driving Steady')],
>   [('delivery_truck', 'In Intersection'), ('box', 'Falling')],
>   [('box', 'On Ground'), ('ego_vehicle', 'Braking')],
>   [('delivery_truck', 'Exiting Intersection'), ('ego_vehicle', 'Stopped')]]
> ```

Scenarios involving the ambulance posed additional challenges due to the constraints imposed by its vehicle dynamics. Specifically, the ambulance's slower acceleration made it difficult to overtake another vehicle within a short distance.

In the manipulation domain, most failures occurred in scenarios requiring objects to be placed inside articulated objects. For example, all scenarios involving the behavior "*the robot opens the table door*" failed because objects could not be placed inside the table closet. Unlike in the driving domain, manipulation scenarios did not involve dynamic actors. Consequently the FSM constraints in manipulation were less complex, avoiding the issues encountered in driving.

However, the manipulation domain presents unique challenges and limitations to manipulation. For instance, scenarios scenarios such as "*rotate lamp head to remove screen glare*" require determining object orientations—an open problem due to the arbitrary canonical orientations of meshes in simulation. Similarly, simulating other modalities like temperature or sound is not currently feasible in PyBullet. For example, tasks such as "*thawing frozen vegetables*" cannot simulate changes in temperature. While our method correctly identifies relevant objects to retrieve, such as "frozen vegetables," these limitations underscore gaps in current simulation capabilities.

| Input Behavior | Scenario Name | Feasible/Infeasible of FSM |
|---|---|---|
| | 1.1) Traffic light malfunction | Feasible |
| | 1.2) Distracted driver late start | Feasible |
| Driving forward from being stationary | 1.3) Intersection congestion | Feasible |
| | 1.4) Pedestrian jaywalking | Feasible |
| | 1.5) Let emergency vehicle pass at intersection | Feasible |
| | 2.1) Traffic congestion due to lane closure | Feasible |
| | 2.2) Letting emergency vehicle pass | Infeasible |
| Slowing down | 2.3) A large truck ahead stopped abruptly | Feasible |
| | 2.4) A vehicle cutting in | Feasible |
| | 2.5) Speed enforcement | Feasible |
| | 3.1) Road assistance | Feasible |
| | 3.2) Elder walking on the street | Feasible |
| Stop abruptly while driving forward | 3.3) Accident ahead | Feasible |
| | 3.4) Yield to ambulance | Feasible |
| | 3.5) Parked car door open | Feasible |
| | 4.1) Protest on the streets | Infeasible |
| | 4.2) Parked car at intersection corner | Feasible |
| Stop abruptly after taking a turn | 4.3) Police checkpoint | Feasible |
| | 4.4) Letting the ambulance pass | Infeasible |
| | 4.5) Picking up a passenger | Feasible |
| | 5.1) Ambulance entering intersection | Feasible |
| | 5.2) Sudden traffic signal change | Feasible |
| Stop abruptly while crossing an intersection | 5.3) Another vehicle running a red light | Feasible |
| | 5.4) Object falling out of truck | Infeasible |
| | 5.5) Police chase | Feasible |
| | 6.1) Debris in front | Feasible |
| | 6.2) Slow traffic | Infeasible |
| Changing lanes while driving forward | 6.3) Yielding for an emergency vehicle | Infeasible |
| | 6.4) Lane closure | Feasible |
| | 6.5) Driver going in the wrong direction | Infeasible |

Table 6: Breakdown of scenario feasibility for driving

| Input Behavior | Scenario Name | Feasible/Infeasible of FSM |
|---|---|---|
| | 1.1) Block unpleasant odor | Feasible |
| | 1.2) Minimize outside noise to watch a movie | Feasible |
| | 1.3) Control ambient lighting | Feasible |
| The robot closes the window | 1.4) Block outside construction site noise | Feasible |
| | 1.5) Avoid prying eyes | Feasible |
| | 1.6) Ensure confidentiality during a private conversation | Infeasible |
| | 1.7) Block direct sunlight to protect indoor plants | Feasible |
| | 2.1) Improve air ventiliation | Feasible |
| The robot opens the door | 2.2) Check delivered package | Infeasible |
| | 2.3) Bid farewell and let the guest out | Feasible |
| | 2.4) Allow a pet to enter | Feasible |
| | 3.1) Bake a cake | Feasible |
| The robot adjusts the oven temperature | 3.2) Roast vegetables | Feasible |
| | 3.3) Preheating the oven | Infeasible |
| | 3.4) Warming a ceramic cup | Infeasible |
| | 4.1) Clean dirty dishes | Feasible |
| | 4.2) Watering plants | Feasible |
| | 4.3) Filling a water bottle | Feasible |
| The robot adjusts the water flow | 4.4) Thawing frozen vegetables | Feasible |
| | 4.5) Washing mixed fruits in a bowl | Feasible |
| | 4.6) Soaking a sponge | Feasible |
| | 4.7) Wash off broccoli in sink | Feasible |
| | 4.8) Cleaning a coffee mug | Feasible |
| | 5.1) Looking for a cereal box | Infeasible |
| The robot open the table doors | 5.2) Showing table content to someone | Infeasible |
| | 5.3) Retrieve a pet toy kept out of sight | Infeasible |
| | 5.4) Look for a hiding pet | Infeasible |
| | 6.1) Clearing the floor after playtime | Feasible |
| The robot is storing an item into storage furniture | 6.2) Put away toys as guest arrive | Feasible |
| | 6.3) Removing choking hazard near pets | Feasible |
| | 6.4) Store detergent out of children's reach | Feasible |
| | 7.1) Retrieve ingredients for cooking | Infeasible |
| The robot is retrieving an item from a fridge | 7.2) Clearing expired items | Infeasible |
| | 7.3) Offering food for a visitor | Feasible |
| | 8.1) Brighten the desk to read a book | Feasible |
| The robot is turning on a lamp | 8.2) Showcase an artwork | Feasible |
| | 8.3) Setting up a small photography shoot to avoid shadows | Feasible |
| | 8.4) Setup working environment | Feasible |
| The robot tilt the display screen | 9.1) Remove glare from display screen | Feasible |
| | 9.2) Sharing a presentation during a meeting | Feasible |
| | 10.1) Searching for an item | Infeasible |
| The robot pull a drawer out | 10.2) Looking for documents | Infeasible |
| | 10.3) Finding a pencil | Infeasible |
| | 11.1) Unpacking boxes | Feasible |
| The robot is retrieving an item from a box | 11.2) Checking content of box | Infeasible |
| | 11.3) Inspection at warehouse | Feasible |
| | 11.4) Retrieving an item for a person | Infeasible |
| | 12.1) Transporting soil for gardening | Infeasible |
| The robot is carrying a bucket | 12.2) Fetching water for mopping the floor | Feasible |
| | 12.3) Grabbing a bucket to fill sand | Feasible |

Table 7: Breakdown of scenario feasibility for manipulation

## A.5   FULL PROMPTS

### A.5.1   NODE PROPOSAL

---
**Prompt: Event Node Proposal Prompt**

```
Input variables: causal_graph
~~~~~~~~~~~~~~~~~~~~~~~~~~~~~~~~~~~~~~~~~~~~~~~~~~
Prompt:
You are an expert in driving scenarios.

In this scenario there is {causal_graph}.

Please provide a list of all the plausible scenarios that caused
    {causal_graph[-1]}. Make sure it is precise. Provide the final answer as a
    comprehensive list of plausible scenarios in the following format within
    the tags <Answer>...</Answer>:

    - cause_name: cause description

Answer: Let's think step by step.
```
---

**Example 7: Event Node Proposal LLM Response**

```
**Input**:
causal_graph = ["ego-vehicle stopping abruptly"]

**Output**:
Sure, lets think through plausible scenarios step by step.
...
<Answer>
- **Pedestrian crossing**: A pedestrian unexpectedly crosses in front of the
    ego-vehicle.
- **Vehicle cutting in**: Another vehicle unexpectedly cuts in front of the
    ego-vehicle.
...
- **Mechanical failure**: A sudden internal failure such as brake malfunction
    leads to abrupt stopping.
- **Emergency vehicle**: An emergency vehicle suddenly appears or signals the
    ego-vehicle to yield.
- **Cyclist lane violation**: A cyclist suddenly moves into the ego-vehicles
    path.</Answer>
```
---

**Prompt: Property Node Proposal Prompt**

```
Input variables: node_name, candidate_values,
~~~~~~~~~~~~~~~~~~~~~~~~~~~~~~~~~~~~~~~~~~~~~~~~~~
Prompt:
You are an expert in driving scenarios.

In this scenario there is {causal_graph}. The entities in the scenario are:
{entities_name}

Please provide a list of all the possible {node_name}s for the entities in the
    scenario, excluding the ego-vehicle. Make sure it is precise. Provide the
    final answer as a comprehensive list of possible {node_name}s in the
    following format within the tags <Answer>...</Answer>:

    - entity_name: ['{node_name}1', '{node_name}2', ...]

Answer: Let's think step by step.
```
---

---

**Example 8: Property Node Proposal LLM Response**

```
**Input**:
node_name = "starting location"
causal_graph = ["emergency vehicle", "ego-vehicle stopping abruptly"]
entities = [{'name': 'ambulance1',
             'type': 'agent',
             'entity_name': 'ambulance',
             'behavioral_properties': {}},
            {'name': 'ego-vehicle',
             'type': 'agent',
             'entity_name': 'ego-vehicle',
             'behavioral_properties': {'action': 'Vehicle drives straight and
     suddenly stops'}}]

**Output**:
Sure, lets think through plausible locations step by step.
...
<Answer>
- ambulance1: ['in the right lane behind the ego-vehicle', 'in the right lane in
     front of the ego-vehicle', 'in the left lane behind the ego-vehicle', 'in
     the left lane in front of the ego-vehicle']</Answer>
```

## A.5.2 Edge Creation

**Prompt: Property Node Proposal Prompt**

```
Input variables: node_name, candidate_values, entities
~~~~~~~~~~~~~~~~~~~~~~~~~~~~~~~~~~~~~~~~~~~~~~~~~~~~
Prompt:
You are an expert in driving scenarios.

The possible {node_name}s for each entity are:
{candidate_values[entity["name"]] for entity in entities}

What are all the possible {node_name}s for each entity in the scenario? To
     answer this, first, please summarize the details of each entities in the
     scenario. Then, check each to see if it is the possible outcome given what
     is known. For each, start by stating everything that is known about all the
     entities, then check if it is plausible given what is known, finally give
     your conclusion. Think step by step. You must not assume additional actions
     beyond what is explicitly described in the behavior. You must also assume
     that the actions are executed fully. Your evaluation needs to be in the
     following format:

1. **Name of {node_name}**
- Known:
- Analysis: (think step by step)
- Contradictions to what is known: (think step by step)
- Conclusion:

Finally, provide the final answer as a list of locations in the following
     formats within the tags <Answer>...</Answer>.
- entity_name: ['{node_name}1', '{node_name}2', ...]

Here are some tips to help you answer the question:
- You may assume that the vehicles can break traffic rules as long as it is
     plausible in real life (realistic). However, the vehicles action must not
     violate the behavior described.
- The {node_name}s selected can only be from the list of possible {node_name}s
     provided.

Answer: Let's think step by step.
```

**Example 9: Property Node Proposal LLM Response**

```
**Input**:
node_name = "starting location"
causal_graph = ["emergency vehicle", "ego-vehicle stopping abruptly on left
     lane"]
entities = [{'name': 'ambulance1',
             'type': 'agent',
             'entity_name': 'ambulance',
             'behavioral_properties': {}},
            {'name': 'ego-vehicle',
             'type': 'agent',
             'entity_name': 'ego-vehicle',
             'behavioral_properties': {'action': 'Vehicle drives straight and
     suddenly stops'}}]
candidate_values = {"ambulance1": ['in the right lane behind the ego-vehicle',
     'in the right lane in front of the ego-vehicle', 'in the left lane behind
     the ego-vehicle', 'in the left lane in front of the ego-vehicle']}

**Output**:
Sure, lets think through plausible locations step by step.
...
<Answer>
- ambulance1: ['in the right lane behind the ego-vehicle']</Answer>
```

