# OpenReview forum: "ReGen: Generative Robot Simulation via Inverse Design"
_ICLR.cc/2025/Conference — ICLR 2025 Poster_

### Official Review · Reviewer_hjAh · 2024-11-02

**Soundness:** 3
**Presentation:** 3
**Contribution:** 2
**Rating:** 6
**Confidence:** 4

**Summary:**

This paper presents ReGen, a generative simulation framework that automates the creation of robot simulation environments through inverse design. The approach leverages large language models (LLMs) to construct and expand graphs capturing cause-and-effect relationships and environmental properties, which are then converted into executable simulation environments.

**Strengths:**

- Appealing core functionality: leveraging inverse design with large language models to generate realistic and diverse simulation environments by capturing cause-and-effect relationships and relevant environmental properties.
- Great experiments to show the strong capability to generate diverse and corner-case of the scenarios.

**Weaknesses:**

1. The paper's novelty analysis is insufficient:
- The causality reasoning and diversity aspects appear to be inherent properties of the LLMs rather than unique contributions of the framework
- The generation capabilities seem heavily dependent on the underlying simulator's features

2. There is no analysis of the generated graph, only the results are shown, making it unclear whether the main contributions come from the LLM rather than the proposed graph structure. Additionally, there is no ablation study on the graph; for instance, it is unclear what would happen without edge construction.

**Questions:**

1. In line 161, what description is used when inferring the property node from the source node?
2. In line 155, is there an example that clarifies what the prior represents exactly?
3. This paper demonstrates a plausible generation ability; however, it lacks analysis of the entire generated graph. Could you provide the complete graph generated by ReGen?
4. Is there an issue with the brackets in line 182? It seems there are multie understandings here. A more precise notation or explanation is needed.
5. When expanding the event-to-event nodes, if the LLM generates events that the simulator cannot execute, how does the system handle this?
6. ReGen exhibits strong capabilities in generating diverse scenarios, yet realism is not thoroughly addressed in the large-scale experiments. Could you elaborate on the realism of the generated scenarios? If the generated scenarios are not reasonable, off-the-shelf VLM understanding may perform poorly, as observed in Table 3.

---

> ### Author Response · Authors · 2024-11-28
> **Pt 1**
>
> We thank reviewer hjAh for their thoughtful feedback and for highlighting areas that could benefit from further explanation. Below, we provide additional experimental studies and discussions to address these concerns.
>
> > The causality reasoning and diversity aspects appear to be inherent properties of the LLMs rather than unique contributions of the framework
>
> - Diversity: Changing models, adjusting hyperparameters, or relying on prompt-engineering methods like ChatScene, which uses GPT-4, result in only marginal improvements in scenario diversity. This is because these methods are limited by the context specified in their prompts, restricting the scope of generated scenarios.  In contrast, our method expands the range of possible scenarios by proposing new, potentially unrelated contexts, which are then validated for plausibility using an LLM as a classifier, resulting in significantly greater diversity (Table 1-2).
> - Causal reasoning: Our framework uses the inherent causality reasoning capability of LLMs to ensure that generated scenarios are realistic. Here, we define a realistic scenario as one in which cause-and-effect relationships are logical—for example, an "emergency vehicle behind the ego vehicle" causing the "ego vehicle to change lanes."
>
> > Generation capabilities seem heavily dependent on the underlying simulator's features
>
> We appreciate the reviewer's observation regarding the dependence on the underlying simulator’s features. However, we view this as a strength of our method: its ability to achieve significantly greater utilization of the simulator’s features compared to prior approaches. For example, as demonstrated in Table 1 and Figure 5, our method achieved greater scenario diversity than ChatScene—which also generates scenarios within the same simulator.
>
> Our results highlight that our method not only adheres to the constraints imposed by the simulator but also leverages its features more effectively to produce a broader range of scenarios. Moreover, as shown in Table 2 and Figure 3, we illustrate that our approach is not confined to a single simulator or domain but is extendable across various simulation platforms.
>
> > There is no analysis of the generated graph, only the results are shown, making it unclear whether the main contributions come from the LLM rather than the proposed graph structure.
>
> Figure 7 presents an ablation study showing the impact of changing event-to-event nodes on scenario diversity. Specifically, the pairwise diversity distribution of generated scenarios is categorized by the proportion of scenarios with unique event-to-event edges: 100%, 80%, 40%, and 0%. A value of 100% indicates that all compared scenarios have completely distinct causes, while 0% means they share the same cause but differ in properties such as start location or behavior.
>
> Notably, the bimodal distribution observed in the results highlights two key effects: introducing new event-to-event nodes significantly enhances diversity by expanding the range of causal relationships, while modifications to property-to-entity nodes result in more nuanced variations. Together, these mechanisms offer greater control over the scope and granularity of the generated scenarios. This analysis has been incorporated into the manuscript in the second paragraph of Appendix A.3.
>
> > Additionally, there is no ablation study on the graph; for instance, it is unclear what would happen without edge construction.
>
> Thank you for your feedback. We conducted an ablation study to evaluate the accuracy of edge creation during the graph expansion stage, with results presented in Table 4 and further detailed in the first paragraph of Appendix A.3.
> - Detail: We assessed event-to-event edges by testing the LLM’s ability to distinguish between causal and non-causal variables. For entity-to-event edges, we evaluated whether the LLM accurately identified simulatable events and selected appropriate assets, consistently demonstrating high performance. Finally, for property-to-entity edges, we tested the LLM’s ability to select the most plausible location, which posed a greater challenge compared to simpler properties, such as determining whether a siren should be on, where it performed more reliably.

---

> ### Author Response · Authors · 2024-11-28
> **Pt 2**
>
> > In line 161, what description is used when inferring the property node from the source node?
>
> Property nodes specify attributes for each entity, including elements such as location, which is dynamically generated using an LLM (e.g. “in the left lane in front of the ego-vehicle”),  or the possible states of an entity in the simulator, which are retrieved from the asset database (see Appendix 1.1–1.2 for additional details). For example, candidate property nodes for a traffic light include all its possible states in the simulator, such as “red,” “green,” “yellow,” and “off.” These proposed nodes serve as candidates for the edge construction step. Examples of this process can be found in Appendix A.1.2. Furthermore, the original manuscript has been updated in the 'Node Proposal' section of the Method to incorporate this clarification.
>
> > In line 155, is there an example that clarifies what the prior represents exactly?
>
> The prior refers to external constraints or preferences that guide the node proposal process. Specifically, the prior can take the form of a word or a list of words derived either from (i) human preferences, (ii) or from the entities available within the simulation environment. For example, if "police car" is an entity in the simulation engine, the prior guides the LLM to generate candidate nodes related to this entity. The corresponding prompt might be, “Keywords for police car.” In response, the LLM could propose candidate nodes such as “police chase.” These candidates are then evaluated during the edge construction step to ensure they align with plausible causal relationships to the source node, e.g., “ego vehicle to stop ← police chase.“
>
> > This paper demonstrates a plausible generation ability; however, it lacks analysis of the entire generated graph. Could you provide the complete graph generated by ReGen?
>
> We appreciate the reviewer’s feedback regarding the need for more comprehensive documentation of our methodology. In response, we provide the complete graph generated by Regen in “Code Example 6” in Appendix A.2.
>
> More detailed examples are provided in Appendix 1.2 for nodes, covering event, entity, and property nodes, and in Appendix 1.3 for edge proposals, including event-to-event, entity-to-event, and property-to-entity connections. Full output examples can also be found in Appendix 5. For code examples, refer to Appendix A1.1 for the asset database, A2 for the finite state machine, and A2.2 for the full scenario configuration example. Additionally, we have also added comprehensive prompt examples for LLMs in Appendix 5. Specifically, Appendix 5.1 contains prompts for node proposals, and Appendix 5.2 includes prompts for edge creation.
>
> > Is there an issue with the brackets in line 182? It seems there are multie understandings here. A more precise notation or explanation is needed.
>
> We acknowledge that our explanation was unclear and have revised this section for clarity.
>
> > When expanding the event-to-event nodes, if the LLM generates events that the simulator cannot execute, how does the system handle this?
>
> Our method employs an LLM as a general classifier, inherently performing rejection sampling by evaluating the plausibility of candidate connections in the context of the source node, thereby ensuring that only simulatable events are generated.
> - Detail: For event-to-event edges, this approach identifies causal relationships. For entity-to-event and property-to-event edges, it evaluates whether the entities (e.g., an ambulance) and their properties (e.g., a siren) are available in the simulation engine. These entities and properties are defined in the asset database (see Appendix 1.1). We have revised the “Event Construction” section of the method to enhance clarity and provide a more detailed explanation. Additional examples are included in Appendix 1.3, and the accuracy of this classifier is presented in Table 4.

---

> ### Author Response · Authors · 2024-11-28
> **Pt 3**
>
> > Could you elaborate on the realism of the generated scenarios? If the generated scenarios are not reasonable, off-the-shelf VLM understanding may perform poorly, as observed in Table 3.
>
> - We agree that realism plays a crucial role in ensuring the generated scenarios align with human expectations and enable better performance for off-the-shelf VLMs. However, we observed that VLMs frequently responded with deceleration as the default action. This behavior, consistent with prior findings [A1], indicates VLMs struggle to reason through nuanced spatial and situational situations. In contrast, scenarios generated by DriveCoT (which also uses the CARLA simulator) involve spawning objects directly in the ego-vehicle’s path, requiring a deceleration response. Here, the VLM's default bias aligns with the expected behavior. The observed failures highlight a limitation of off-the-shelf VLMs in reasoning about complex driving situations, rather than pointing to an inherent lack of realism in the scenario design. Importantly, this limitation cannot be attributed to insufficient input, as the VLM is provided with privileged information, including the location and speed of other cars.
>
> - We conducted a preliminary human study to evaluate the realism of the generated simulations. In this study, participants assess whether the scenarios are both realistic and consistent with their descriptions. We will include the analysis of the results from the study in a future revision.
>
> ---
>
> A1. S. Sreeram, T.-H. Wang, A. Maalouf, G. Rosman, S. Karaman, and D. Rus, “Probing Multimodal LLMs as World Models for Driving,” arXiv.org, 2024. https://arxiv.org/abs/2405.05956.

---

> > ### Author Response · Authors · 2024-11-29
> > **Could you please let us know your feedback**
> >
> > Dear Reviewer hjAh,
> >
> > Thank you once again for dedicating your time and effort to reviewing our paper. We have worked diligently to address all your concerns, including the revised manuscript and additional experiments to further strengthen our work. Could you kindly let us know if you have any further questions or concerns?
> >
> > Much appreciated,
> >
> > Best regards,
> >
> > Authors.

---

> > > ### Author Response · Authors · 2024-12-01
> > > **2 Days Left for Rebuttal -- We would like to know your feedback**
> > >
> > > Dear Reviewer hjAh,
> > >
> > > We wanted to follow up on our earlier response to your feedback.
> > >
> > > Thank you once again for dedicating your time and effort to reviewing our paper. We have worked diligently to address all your concerns, including revising the manuscript and providing additional experiments to further strengthen our work.
> > >
> > > With only two days remaining in the rebuttal period, we would greatly appreciate it if you could let us know if there are any remaining questions or concerns we can address.
> > >
> > > We look forward to hearing from you. Thank you!
> > >
> > > Best regards,
> > >
> > > Authors

---

> > > > ### Author Response · Authors · 2024-12-02
> > > > **Could you please give us feedback**
> > > >
> > > > Dear Reviewer hjAh,
> > > >
> > > > We greatly appreciate your thoughtful feedback, which has been invaluable in helping us improve the quality and clarity of our paper.
> > > >
> > > > We are encouraged that our rebuttal has successfully addressed the concerns raised by the other reviewers, and we hope it has similarly clarified any questions or concerns you may have. As the rebuttal period comes to an end, we kindly ask for your thoughts on our responses and invite you to consider revisiting your evaluation. If you feel your concerns have been resolved, we would be grateful if you could raise your score accordingly. If there are any remaining points that need clarification, please do not hesitate to let us know, and we will gladly address them promptly.
> > > >
> > > > Thank you again for your time.
> > > >
> > > > Best regards,
> > > >
> > > > Authors

---

### Official Review · Reviewer_x8VP · 2024-11-03

**Soundness:** 2
**Presentation:** 2
**Contribution:** 2
**Rating:** 3
**Confidence:** 4

**Summary:**

This paper introduces ReGen, a generative simulation framework that automates the creation of robot simulation environments using inverse design. ReGen takes a robot's behavior and textual descriptions as input, then uses large language models to construct and expand causal graphs that capture relationships between events, entities, and their properties, which are then converted into executable simulation environments. The framework is implemented and validated in both autonomous driving and robot manipulation tasks, demonstrating capabilities in augmenting simulations based on ego-agent behaviors, generating counterfactual scenarios, reasoning about agent cognition, and handling different sensing modalities. The experimental results show that ReGen generates more diverse environments compared to existing simulations, effectively creates corner cases for safety-critical applications, and produces more challenging vision-language-action datasets for vision language models.

**Strengths:**

While I do not see a clear strength in terms of novelty. However, the authors provide a comprehensive evaluation against multiple baselines, and thorough implementation across autonomous driving and manipulation tasks to demonstrate the applicability of the proposed method.

**Weaknesses:**

The main contribution of this paper is vague. The use of LLMs to generate simulated task environments for robotic tasks cannot be considered as a major contribution. The proposed method of using LLMs for graph expansion and simulation generation appears to be a straightforward extension of existing work such as [a, b]. I do not see a convincing argument that states the technical contributions of the proposed approach other than some prompt engineering. The "inverse design" is essentially a rebranding of standard goal-conditional generation approaches, where the goal is guided by the inverse design and feed into the LLMs.

The evaluation of the proposed method is not thorough enough, with only brief mentions of success rates without analysis of failure modes, no comparison with human-designed scenarios, and unclear metrics for scenario complexity beyond embedding diversity. Furthermore, the paper inadequately addresses fundamental simulation constraints, as many proposed scenarios can't be fully simulated due to engine limitations, but there is no systematic approach for dealing with these limitations or assessing their impact on practical utility.

[a] Wang, Yufei, et al. "Robogen: Towards unleashing infinite data for automated robot learning via generative simulation." arXiv preprint arXiv:2311.01455 (2023).
[b] Zhang, Jiawei, Chejian Xu, and Bo Li. "ChatScene: Knowledge-based safety-critical scenario generation for autonomous vehicles." In Proceedings of the IEEE/CVF Conference on Computer Vision and Pattern Recognition. 2024.

------

Additional comments:

Given that direct responses are no longer possible on the original [thread](https://openreview.net/forum?id=EbCUbPZjM1&noteId=yeUwITDALH), I am sharing my reply here.

Training policies in complex environments demands significantly greater computational resources and effort (e.g., see [c]). The fact that your policy was obtained in just 2 minutes suggests the task and environment were overly simplified comparing to actual scenarios. I have concerns about whether your proposed method would remain effective when scaled to more complex tasks and environments.

Regarding your comments about the primary contribution, I remain unconvinced. The work does not demonstrate any novel or innovative applications of LLMs. Therefore, I will keep my original rating.

[c] Gu, Jiayuan, et al. "Multi-skill mobile manipulation for object rearrangement." arXiv preprint arXiv:2209.02778 (2022).

**Questions:**

1. Regarding the LLM implementation: Could you provide more details about the specific prompting strategies used for graph expansion? How do you ensure consistency and reliability in the LLM output, and how do you handle cases where the LLM generates invalid or inconsistent relationships?
2. Regarding the failures: The paper reports success rates of 80% for driving and 78% for manipulation. Could you provide a detailed analysis of the failure modes?
3. About simulation constraints: How do you systematically identify and handle cases where desired scenarios exceed the capabilities of the simulation engine? Is there a formal process for determining which aspects of a scenario can be reasonably approximated and which must be excluded?
4. About generated task completeness. How do you guarantee that the generated tasks are feasible for a robot or autonomous agent? i.e., that there is a practical solution to the generated tasks.

---

> ### Author Response · Authors · 2024-11-28
> **Pt 1**
>
> We thank reviewer x8VP for their thoughtful feedback and for highlighting areas that could benefit from further explanation. Below, we provide additional experimental studies and discussions to address these concerns.
>
> > Do not see a clear strength in terms of novelty
>
> We appreciate the reviewer’s feedback and recognize the need to better articulate the novelty of our approach.
> - Previous methods [A1, A2] aim to increase the diversity of simulations by generating new reward functions, which in turn produce more varied behaviors. This process is computationally intensive and inherently constrained by the need to define distinct reward functions for every simulated scenario.
> - Our method proposes reusing and adapting existing behaviors (e.g., a trajectory or reward function) to generate novel simulations. Our method achieves this by altering the environmental context in which these behaviors occur. This approach overcomes the bottleneck of prior methods and enables the creation of richer and more varied scenarios, as extensively demonstrated through experiments. The key insight is that while behaviors are inherently limited, the environments in which they occur are far more diverse. For example, the abrupt stopping of a self-driving car can apply to various contexts, such as encountering a red traffic light, responding to a pedestrian stepping into the road, or yielding to an approaching police car with its siren on.
>
> > No comparison with human-designed scenarios
>
> One of the methods we compared against is DriveCoT, which employs a rule-based expert policy to generate ground truth labels for driving scenarios selected from the CARLA Leaderboard 2.0, based on the NHTSA crash typology. We consider these scenarios to be designed by human experts.
> First, we evaluated the diversity of scenarios generated by our method compared to those from DriveCoT, with our method achieving a significantly higher diversity score (Table 1). Second, we assessed the performance of VLMs on scenarios from both DriveCoT and our method (Table 3).
>
> > Unclear metrics for scenario complexity beyond embedding diversity
>
> In our study, we measure complexity by assessing how challenging it is for VLMs to respond accurately to a scenario (Table 3). By evaluating VLM performance in these scenarios, we aim to provide a practical metric for assessing scenario complexity beyond embedding diversity. For example, we observed that in lane-change scenarios such as "avoiding debris," "overtaking a slow vehicle," "merging into an open lane," or "swerving to avoid a wrong-way driver," off-the-shelf VLMs often defaulted to deceleration as the primary action. This behavior, consistent with prior findings [A3], indicates a struggle to reason through nuanced spatial and situational situations. In contrast, scenarios generated by methods like DriveCoT (which also uses the CARLA simulator) involve spawning objects directly in the ego-vehicle’s path, prompting straightforward deceleration responses. Here, the VLM's default bias aligns with expected behaviors, demonstrating a lower level of complexity.
>
> > The paper inadequately addresses fundamental simulation constraints, as many proposed scenarios can't be fully simulated due to engine limitations, but there is no systematic approach for dealing with these limitations or assessing their impact on practical utility.
>
> In our approach, a scenario is represented as a graph, with nodes representing events, entities, and properties, and edges capturing transitions between them. To ensure plausibility, we employ an LLM as a classifier to evaluate the validity of each edge. We have revised the Methodology section to improve clarity. To address simulation constraints explicitly, we construct an asset database represented as a directed graph, which encodes the capabilities of the simulation engine (see Appendix 1.1).
>
> ---
>
> A1. Zhang, Jiawei, et al. “ChatScene: Knowledge-Enabled Safety-Critical Scenario Generation for Autonomous Vehicles.” ArXiv.org, 2024, arxiv.org/abs/2405.14062.
>
> A2. Wang, Yufei, et al. “RoboGen: Towards Unleashing Infinite Data for Automated Robot Learning via Generative Simulation.” ArXiv.org, 2 Nov. 2023, arxiv.org/abs/2311.01455.
>
> A3. S. Sreeram, T.-H. Wang, A. Maalouf, G. Rosman, S. Karaman, and D. Rus, “Probing Multimodal LLMs as World Models for Driving,” arXiv.org, 2024. https://arxiv.org/abs/2405.05956.

---

> ### Author Response · Authors · 2024-11-28
> **Pt 2**
>
> > Could you provide more details about the specific prompting strategies used for graph expansion?
>
> Thank you for your feedback. We have added comprehensive prompt examples for LLMs in Appendix 5. Specifically, Appendix 5.1 contains prompts for node proposals, and Appendix 5.2 includes prompts for edge creation.
> Additionally, detailed examples are provided in Appendix 1.2 for nodes, covering event, entity, and property nodes, and in Appendix 1.3 for edge proposals, including event-to-event, entity-to-event, and property-to-entity connections. Full output examples can also be found in Appendix 5. For code examples, refer to Appendix A1.1 for the asset database, A2 for the finite state machine and the full scenario configuration example.
>
> > How do you ensure consistency and reliability in the LLM output, and how do you handle cases where the LLM generates invalid or inconsistent relationships?
>
> We set both the temperature and top-p to 0 to increase the consistency in the LLM output, which empirically yields more deterministic results. Our ablation experiment (Table 4) demonstrated minimal variations in its responses.
>
> > Could you provide a detailed analysis of the failure modes
>
> Thank you for raising this point. We have updated the manuscript to include a detailed analysis of the failure modes observed in two key stages of our method: (1) the graph expansion stage (Appendix A.3, paragraph 1) and (2) the grounding into simulation (Appendix A.4.2).
>
> > About generated task completeness. How do you guarantee that the generated tasks are feasible for a robot or autonomous agent? i.e., that there is a practical solution to the generated tasks
>
> Our approach ensures task feasibility by converting each scenario generated from the graph expansion process into a finite state machine (FSM). The FSM defines a satisfiability problem, which we solve using tools like Google’s CP-SAT solver. The solver finds solutions for the variables such as the x, y coordinates of the start and end positions $(x_0, y_0, x_T , y_T)$, as well as the speed, such that it satisfies the constraints imposed by the FSM. There is a practical solution if the simulation terminates in a state that satisfies the terminal condition of the FSM. The feasibility rate for driving is 80% and manipulation is 78%.
>
> - Example: The FSM shown below is for the scenario “Ego-vehicle stopping ← Ambulance approaching from behind.” The FSM translates low-level trajectory into abstract states for state tracking. For example, the abstract state “Ambulance Approaching” is defined as a constraint of where the ambulance is behind the ego-vehicle and also in motion `if behind_ego(“ambulance”) and is_currently_moving(“ambulance”)`. Additional details on this process can be found in Appendix A.2
>
> ```
> fsm = [[(’ambulance1’, ’Ambulance Approaching’), (’ego-vehicle’, ’Ego Driving Steady’)],
>  [(’ambulance1’, ’Ambulance Close to Ego’)],
> [(’ego-vehicle’, ’Ego Braking’)],
> [(’ego-vehicle’, ’Ego Stopped Abruptly’)],
> [(’ambulance1’, ’Ambulance Passing Ego’)]]
> ```

---

> > ### Author Response · Authors · 2024-11-29
> > **Please let us know your feedback**
> >
> > Dear Reviewer x8VP:
> >
> > We sincerely appreciate your insightful comments and advice, which have been instrumental in improving the quality and clarity of our paper.
> >
> > We hope that the revisions, along with the additional details and experimental results we provided, have sufficiently addressed your concerns. As the rebuttal period nears its conclusion, could you kindly let us know if there are any remaining questions or concerns?
> >
> > Best,
> >
> > Authors

---

> > > ### Author Response · Authors · 2024-12-01
> > > **Rebuttal Deadline in 2 Days -- Could you please let us know your feedback**
> > >
> > > Dear Reviewer x8VP,
> > >
> > > We wanted to follow up on our earlier response to your feedback on our manuscript.
> > >
> > > We sincerely appreciate the time and effort you have dedicated to reviewing our work. As mentioned, we have made revisions and included additional details and experimental results to address your concerns. With only two days remaining in the rebuttal period, we kindly ask if there are any remaining questions or points you would like us to address.
> > >
> > > We hope to hear from you soon. Thank you!
> > >
> > > Best regards,
> > > Authors

---

> > > > ### Author Response · Authors · 2024-12-02
> > > > **Rebuttal Period Closing Soon – Could you let us know your feedback**
> > > >
> > > > Dear Reviewer x8VP,
> > > >
> > > > We sincerely appreciate your valuable feedback, which has greatly helped us enhance the quality and clarity of our paper.
> > > >
> > > > We are encouraged that our rebuttal has effectively addressed the concerns raised by the other reviewers, and we hope it has similarly clarified any questions you may have. As the rebuttal period draws to a close, we kindly request your thoughts on our rebuttal and ask that you consider raising your score if you feel your concerns have been resolved. If there are any remaining issues, please do not hesitate to let us know, and we will do our best to address them promptly.
> > > >
> > > > Best regards,
> > > >
> > > > Authors

---

> > > > > ### Author Response · Authors · 2024-12-03
> > > > > **Rebuttal Ending Soon - Could you let us know your feedback and revisit evaluation**
> > > > >
> > > > > Dear Reviewer x8VP,
> > > > >
> > > > > Thank you for your valuable feedback, which has significantly contributed to improving the clarity and quality of our paper.
> > > > >
> > > > > We are pleased that our rebuttal has addressed the concerns raised by all other reviewers and hope it has similarly resolved any questions you may have. Could you let us know your feedback and consider revisiting your evaluation if you feel your concerns have been adequately addressed? Thank you!
> > > > >
> > > > > Best regards,
> > > > >
> > > > > Authors

---

> > > > > > ### Author Response · Authors · 2024-12-03
> > > > > > **Final Hours for Rebuttal – Your Feedback Would Be Greatly Appreciated**
> > > > > >
> > > > > > Dear Reviewer x8VP,
> > > > > >
> > > > > > We are pleased to note that our rebuttal has effectively addressed the concerns of all other reviewers, and we are hopeful it has also resolved your queries. If there are any outstanding points from your initial review, we would greatly appreciate it if you could let us know so that we can address them promptly. Otherwise, we kindly request you to consider revisiting your evaluation if you feel our response has adequately resolved your concerns.
> > > > > >
> > > > > > Thank you for your time and thoughtful consideration.
> > > > > >
> > > > > > Warm regards,
> > > > > >
> > > > > > Authors

---

> > > > > > > ### Comment · Reviewer_x8VP · 2024-12-03
> > > > > > > **Thanks for your active and detailed response.**
> > > > > > >
> > > > > > > Dear Authors,
> > > > > > >
> > > > > > > Thank you for your detailed response to the reviewers' comments. I apologize for my delayed reply due to recent illness.
> > > > > > >
> > > > > > > After carefully reviewing your responses to all reviewers, I maintain significant concerns about the fundamental aspects of your work. While you propose using LLMs to validate the feasibility of generated environments, this approach raises several critical questions: 1. How can LLMs effectively validate robot-environment interactions without a in-depth understanding of robot embodiment? Specifically, how can they verify whether objects are reachable within the robot's configuration space? 2. What mechanisms ensure that the generated environments are complete and allow for successful task execution by the robot? These concerns point to a broader issue: the lack of a comprehensive validation protocal for the generated environments. Without validation across multiple dimensions (physical feasibility, task completeness, etc.), it becomes impossible to differentiate between failures caused by the robot model versus environmental constraints. Your current implementation, as shown in Figure 3, presents relatively simple manipulation environments that fall short when compared to the complexity and realism offered by existing benchmarks such as ManiSkill, Habitat, and AI2-THOR. These  environments demonstrate the level of sophistication required for meaningful robot simulation research.
> > > > > > >
> > > > > > > Given these unresolved fundamental issues, particularly regarding environment validation and complexity, I maintain my original rating.

---

> ### Author Response · Authors · 2024-12-03
> **Response to Reviewer**
>
> Dear Reviewer x8VP,
>
> Thank you for your feedback, we appreciate you getting back to us.
>
> > 1. How can LLMs effectively validate robot-environment interactions without a in-depth understanding of robot embodiment? Specifically, how can they verify whether objects are reachable within the robot's configuration space?
>
> We would like to clarify that, in our approach, each simulated environment includes a policy for the ego agent to solve and this takes into account task completeness, physical feasability, and object reachability. We use an LLM to propose high-level constraints -- for example, in the scenario "yielding to an ambulance," the LLM propose a constraint that the ambulance should initially be behind the ego-vehicle before passing ahead -- we do not rely on LLMs to verify whether objects are reachable within the robot's configuration space. Our success rate for task completeness for driving is 80% and manipulation is 78%. We have also provided a detailed analysis of failure modes in Appendix A.4.2.
>
> > 2. What mechanisms ensure that the generated environments are complete and allow for successful task execution by the robot? These concerns point to a broader issue: the lack of a comprehensive validation protocal for the generated environments.
>
> Our setup is slightly different in that we adopt an inverse design approach: given an ego agent behavior, we generate simulated environments where such behaviors could plausibly occur. A key bottleneck in existing methods lies in obtaining effective policies. Our work demonstrates how reusing existing behaviors can enhance the diversity and realism of simulated environments.
>
> > Without validation across multiple dimensions (physical feasibility, task completeness, etc.), it becomes impossible to differentiate between failures caused by the robot model versus environmental constraints. Your current implementation, as shown in Figure 3, presents relatively simple manipulation environments that fall short when compared to the complexity and realism offered by existing benchmarks such as ManiSkill, Habitat, and AI2-THOR. These environments demonstrate the level of sophistication required for meaningful robot simulation research.
>
> While our Figure 3 depicts simpler manipulation scenarios, we also address complex and realistic environments in other domains. For example, in autonomous driving, our generated environments include challenging multi-agent interactions such as "yielding to an emergency vehicle" and "picking up a passenger on the sidewalk," which prior methods [A1, A2, A3] did not address.
>
> For manipulation, we generate complex tasks that involve multiple subtasks. For instance, "storing detergent out of children's reach" requires a sequence of actions: "opening storage furniture," "placing the detergent inside," and "closing the furniture door." Additional examples are available on our website (https://sites.google.com/view/regen-simulation).
>
> Furthermore, our method complements existing methods like ManiSkill, Habitat, and AI2-THOR by enabling the reuse of learned policies in diverse simulated environments. This reusability enhances the diversity and applicability of these policies, allowing for the generation of a broader range of simulated environment (Table 1-2, Figure 5).
>
> Thank you again for your time, and we hope you are feeling better and in good health.
>
> Best,
>
> Authors
>
> ---
> A1. Zhang, Jiawei, et al. “ChatScene: Knowledge-Enabled Safety-Critical Scenario Generation for Autonomous Vehicles.” ArXiv.org, 2024, arxiv.org/abs/2405.14062.
>
> A2. Wang, Yufei, et al. “RoboGen: Towards Unleashing Infinite Data for Automated Robot Learning via Generative Simulation.” ArXiv.org, 2 Nov. 2023, arxiv.org/abs/2311.01455.
>
> A3. S. Sreeram, T.-H. Wang, A. Maalouf, G. Rosman, S. Karaman, and D. Rus, “Probing Multimodal LLMs as World Models for Driving,” arXiv.org, 2024. https://arxiv.org/abs/2405.05956.

---

> ### Comment · Reviewer_x8VP · 2024-12-03
>
> > In Q1 ... "each simulated environment includes a policy for the ego agent to solve and this takes into account task completeness, physical feasability, and object reachability." ...
>
> How did you obtain the policy? Replicate the agent behavior?
>
> > In Q2 ... "Our setup is slightly different in that we adopt an inverse design approach" ...
>
> Your answer does not address this problem. The thing is that the problem raise from such inverse design, so how the inverse design could resolve the problem.
>
> > ... "our method complements existing methods like ManiSkill, Habitat, and AI2-THOR by enabling the reuse of learned policies in diverse simulated environments." ...
>
> Table 1-2 does not demonstrate the proposed method could complements existing benchmarks. More expriment should be conducted to show evidances that the proposed method how handle complex scenes (e.g., a layout with several rooms, not an "table-top" scene). The complexity in domestic scene is quite different from complexity in automonous driving scenes, the latter is not senstive to arrangements of objects while you altering the environment.

---

> > ### Author Response · Authors · 2024-12-03
> > **Response to reviewer x8VP**
> >
> > Dear Reviewer x8VP,
> >
> > Thanks for your prompt responses.
> >
> > > How did you obtain the policy? Replicate the agent behavior?
> >
> > We define behaviors either as a trajectory or a reward function. In driving scenarios, we reuse the trajectory. For manipulation tasks, we retrain a new policy based on the reward function
> >
> > > The thing is that the problem raise from such inverse design, so how the inverse design could resolve the problem.
> >
> > The best validation we can propose is to obtain a policy and test whether it performs effectively in the simulated environment. In such a setup, the validation that determines the correctness of generated simulated environment is to check if the generated environment indeed provides a scenario where the given behavior should have occurred. For driving tasks, we reuse a motion trajectory, while for manipulation tasks, we retrain a new policy using the defined reward function. We would greatly appreciate your thoughts on whether there are alternative validation protocols or methodologies that could further strengthen this evaluation.
> >
> > > Table 1-2 does not demonstrate the proposed method could complements existing benchmarks.
> >
> > Regarding complementing existing benchmarks, our experiments are effectively complementing Robogen. We evaluate our method by comparing it to RoboGen, using a subset of its behaviors (i.e., reward functions) that have already been benchmarked against other baselines. As demonstrated in Table 2, our method achieves greater diversity compared to both the subset we utilized and the full set of RoboGen behaviors.
> >
> > Best,
> >
> > Authors

---

> > > ### Comment · Reviewer_x8VP · 2024-12-03
> > >
> > > Thank you for your response.
> > >
> > > Training a policy usually is time consuming. What is the actual time complexity for validating each newly generated environment (e.g., validation time in minutes or hours per environment)?
> > >
> > > Furthermore, the method's capability to handle complex domestic environments remains questionable. Given that the primary contribution appears to be application-focused rather than theoretical, more thorough experimental evidence is expected to demonstrate that the proposed method indeed significantly improves upon prior arts. However, substantial improvements typically stem from profound insights, which are absent in this paper.
> > >
> > > As it stands, the work risks being categorized as primarily LLM prompt engineering - an increasingly common approach that, while useful, may not meet the threshold for scientific/theoretical contribution in this field.

---

> > > > ### Author Response · Authors · 2024-12-03
> > > > **Response to reviewer x8VP**
> > > >
> > > > Dear Reviewer x8VP,
> > > >
> > > > > Training a policy usually is time consuming. What is the actual time complexity for validating each newly generated environment (e.g., validation time in minutes or hours per environment)?
> > > >
> > > > For manipulation tasks, validation typically takes less than two minutes. For driving scenarios, we empirically observed that it takes approximately 5-15 minutes per environment. This is primarily because each validation requires running a simulation, which takes ~0.5 seconds using CARLA. Notably, in our proposed problem setting, the trajectory of the ego-vehicle is already known, offering opportunities for optimization. However, these optimizations were not implemented in the current work and remain part of our planned future improvements
> > > >
> > > > > As it stands, the work risks being categorized as primarily LLM prompt engineering - an increasingly common approach that, while useful, may not meet the threshold for scientific/theoretical contribution in this field.
> > > >
> > > > The primary role of the LLM in our method is to generate diverse, plausible scenarios. Changing models, adjusting hyperparameters, or relying on prompt-engineering methods like ChatScene, which uses GPT-4, result in only marginal improvements in scenario diversity. This is because these methods are limited by the context specified in their prompts, restricting the scope of generated scenarios. In contrast, our method expands the range of possible scenarios by proposing new, potentially unrelated contexts, which are then validated for plausibility using an LLM as a classifier, resulting in significantly greater diversity (Table 1-2), enabling us to simulate corner-case scenarios for safety-critical applications.
> > > >
> > > > Other reviewers, hjAh and HrM2, raised similar concerns but ultimately expressed that their concerns were thoroughly addressed in the rebuttal and appropriately reflected in the revised manuscript.
> > > >
> > > > Best,
> > > >
> > > > Authors.

---

### Official Review · Reviewer_HrM2 · 2024-11-03

**Soundness:** 2
**Presentation:** 2
**Contribution:** 2
**Rating:** 6
**Confidence:** 4

**Summary:**

This paper presents ReGen, which, given an existing agent's behavior and motion trajectory, generates simulated environments counterfacturally that explains the possible causes and preconditions of this agent trajectory via LLM and VLM. In this way, authors are able to augment simulation data that allows the training of more robust models and policies. The authors then compare their approach to previous baselines in self driving (Carla simulator) and in robot manipulation (PyBullet), demonstrating that their approach effectively generates more diverse simulation environments.

**Strengths:**

Diversifying and augmenting existing simulated environments and trajectories through counterfactural generation is an important research problem. It provides a promosing way to improve the robustness and safety of agents and policies under corner cases and unexpected scenarios.

**Weaknesses:**

The paper suffers from significant weaknesses in writing clarity and the depth of analysis, specifically:
- The methodology section 2 does not detail the models, the model versions, and the important hyperparameters used for prompting LLM / VLM for counterfactual generation. Some of the details are deferred to page 9, which should have been in much earlier in the paper.
- No prompt examples for LLM / VLM are provided in the appendix, making the method unreproducible. No concrete, detailed examples for the entire process of constructing Carla and PyBullet environments through counterfactual generation (following algorithm 1) are provided.
- Algorithm 1 will very likely result in infinite loop, as there will likely always be at least 1 node with input degree < 1 (unless there are circular cause and effect dependencies).
- L320 - ReGen is able to generate simulated environments by augmenting visual observations like GPS measurement jamming. However there is no detailed description of how the visual observations are augmented. It seems that some tool use ability of LLM is invoked.
- The experiments only demonstrate that ReGen outperforms baselines, but **why** ReGen outperforms the baseline is unclear. There are no ablations in e.g., prompting to answer this question. For example, why does ReGen outperform ChatScene? Is it due to better prompting and / or the better VLM used (note that no prompts are provided throughout the paper)? Is the comparison with baseline fair, with the same specific versions of GPT4o / GPT4 / Claude 3.5?

Edit: Thanks authors for the rebuttal! The revision has significantly improved the quality of the paper and I'm increasing my rating.

**Questions:**

Please address all weaknesses listed above.

Minor:
- Fig 4 caption: "desipte" -> "despite"
- "table {x}" should be "Table x"; "figure {x}" should be "Figure {x}"

---

> ### Author Response · Authors · 2024-11-28
>
> We thank reviewer HrM2 for their thoughtful feedback and for highlighting areas that could benefit from further explanation. Below, we provide additional experimental studies and discussions to address these concerns.
>
> > Additional method details
>
> We have clarified that all LLM queries in our method utilize the GPT-4o-2024-08-06 model with a temperature and top-p value of 0. Additionally, we have revised the Methodology section to provide this information at the end of the first paragraph in subsection 2.2.
> More detailed examples are provided in Appendix 1.2 for nodes, covering event, entity, and property nodes, and in Appendix 1.3 for edge proposals, including event-to-event, entity-to-event, and property-to-entity connections. Full output examples can also be found in Appendix 5. For code examples, refer to Appendix A1.1 for the asset database, A2 for the finite state machine and full scenario configuration example.
>
> > Prompt examples
>
> Thank you for your feedback. We have added comprehensive prompt examples for LLMs in Appendix 5. Specifically, Appendix 5.1 contains prompts for node proposals, and Appendix 5.2 includes prompts for edge creation.
>
> > Algorithm 1 will very likely result in infinite loop, as there will likely always be at least 1 node with input degree < 1 (unless there are circular cause and effect dependencies).
>
> The potential for an infinite loop arises because the main graph expansion continuously adds new cause-and-effect relationships. However, this process is controlled by user-defined stopping conditions (graph depth or maximum number of nodes). We have clarified this in the manuscript to improve clarity.
>
> > L320 - ReGen is able to generate simulated environments by augmenting visual observations like GPS measurement jamming. However there is no detailed description of how the visual observations are augmented. It seems that some tool use ability of LLM is invoked.
>
> In the case of GNSS jamming, the entity is the GNSS sensor, and the property is noise. We leverage an LLM to generate a function that invokes add_gnss_noise from the CARLA API, simulating a "GPS Jamming" scenario. The FSM tracks state transitions, enabling us to identify the appropriate moments during the simulation to call this function. This entire process is fully automated. We have revised the original manuscript to clarify this  process and and included examples in Appendix 2 for further illustration.
>
> > Why ReGen outperform the baselines
>
> Our method surpasses ChatScene by significantly diversifying the underlying causes for challenging scenarios. ChatScene primarily generates scenarios involving objects crossing in front of the ego-vehicle (e.g., a pedestrian crossing or a car merging), while our method introduces more complex variations, such as a group of pedestrians or a vehicle traveling in the opposite direction. These scenarios empirically were more challenging for the driving policies, such as requiring larger steering adjustments to avoid groups of pedestrians. We have updated the original manuscript to elaborate on this.
>
> Changing models, adjusting hyperparameters, or relying on prompt-engineering methods like ChatScene, which uses GPT-4, result in only marginal improvements in scenario diversity. This is because these methods are limited by the context specified in their prompts, restricting the scope of generated scenarios.  In contrast, our method expands the range of possible scenarios by proposing new, potentially unrelated contexts, which are then validated for plausibility using an LLM as a classifier, resulting in significantly greater diversity (Table 1-2).
>
> - Detail: Our approach decouples node proposals from edge construction. A key perspective is that graph expansion systematically explores the LLM's knowledge space, uncovering and organizing latent information into a structured graph. The node proposal component in our method generates new context condition on priors. For example, the prior guides the LLM to generate candidate nodes related to an entity such as a “police car”. The corresponding prompt might be, “Keywords for police car.” In response, the LLM could propose candidate nodes such as “police chase.” Then, during edge construction,  the LLM – which acts as a general classifier – evaluates which nodes are plausible within the context to ensure that the resulting scenarios are not only diverse but also simulatable. Together, the node proposal and edge construction steps enable us to explore all potential events and uncover corner-case scenarios.

---

> > ### Author Response · Authors · 2024-11-29
> > **Thank you for raising the score**
> >
> > Dear reviewer HrM2,
> >
> > Thank you very much for raising the score and for acknowledging our efforts during the rebuttal to address your concerns. Your feedback has been invaluable in guiding us to significantly enhance the quality of the paper. We will incorporate the revisions into the final version.
> >
> > If you have any additional comments or suggestions, please feel free to let us know.
> >
> > Best regards,
> >
> > Authors

---

> > > ### Author Response · Authors · 2024-12-01
> > > **Rebuttal Period Closing Soon – Final Suggestions?**
> > >
> > > Dear Reviewer HrM2,
> > >
> > > We wanted to follow up on our earlier note to thank you once again for raising the score and for acknowledging our efforts to address your concerns during the rebuttal process.
> > >
> > > We would greatly appreciate it if you could let us know if you have any additional comments or suggestions, especially as the rebuttal period nears its conclusion.
> > >
> > > We look forward to hearing from you.
> > >
> > > Best regards,
> > >
> > > Authors

---

### Official Review · Reviewer_q9yg · 2024-11-04

**Soundness:** 2
**Presentation:** 2
**Contribution:** 2
**Rating:** 6
**Confidence:** 5

**Summary:**

This paper introduces ReGen, a generative simulation framework that automates the process of constructing simulations using inverse design.  Given an agent’s behavior (such as a motion trajectory or objective function) and its textual description, ReGen infers the underlying scenarios and environments that could have caused the behavior.  This approach leverages large language models (LLMs) to construct and expand a graph that captures cause-and-effect relationships and relevant entities with properties in the environment, which is then processed to configure a robot simulation environment.  ReGen is demonstrated in autonomous driving and robot manipulation tasks, generating more diverse and complex simulated environments compared to existing simulations with high success rates, and enabling controllable generation for corner cases.

**Strengths:**

1. ReGen presents an inverse design approach for generative simulation, which allows for the creation of diverse and complex simulated environments based on agent behavior and textual descriptions.
2. ReGen generates more diverse and complex simulated environments compared to existing simulations, as demonstrated in autonomous driving and robot manipulation tasks.
3. ReGen enables controllable generation for corner cases, which is important for safety-critical applications like autonomous driving.

**Weaknesses:**

1. The evaluation of ReGen is primarily focused on diversity and complexity, with less emphasis on the realism and physical accuracy of the generated simulations.
2. ReGen heavily relies on LLMs, which can be computationally expensive and may not always generate semantically accurate or physically plausible scenarios due to LLMs.
3. The applicability of ReGen to other robotics domains beyond autonomous driving and robot manipulation is not extensively explored.

**Questions:**

See weakness.

---

> ### Author Response · Authors · 2024-11-28
>
> We thank reviewer q9yg for their feedback and for highlighting areas that could benefit from further explanation. Below, we provide additional experimental studies and discussions to address these concerns.
>
> > Primarily focused on diversity and complexity, with less emphasis on the realism and physical accuracy of the generated simulations
>
> Thank you for your feedback. We agree that realism plays a crucial role in ensuring the generated scenarios align with human expectations. We address realism through two aspects.
>
> - First, we ensure the scenario’s realism by validating the plausibility of cause-and-effect relationships between events. For example, a scenario is realistic only if the cause-and-effect is logical, such as an event "emergency vehicle behind the ego-vehicle" causing the event "ego-vehicle to change lanes.”
>
> - Second, we conducted a preliminary human study to evaluate the physical realism of the generated simulations. In this study, participants assess whether the scenarios are both realistic and consistent with their descriptions. We will include the analysis of the results from the study in a future revision.
>
> > LLMs can be computationally expensive and may not always generate semantically accurate or physically plausible scenarios
> Thank you. We address this concern in two parts:
> - Empirically, our method takes approximately 1-3 minutes to generate a scenario using GPT-4o-2024-08-06.
>     - Details: We classify multiple candidates in a single prompt, reducing the query complexity in each iteration of the graph expansion step to $O(1)$. This approach avoids the $O(n^2)$ complexity of pairwise comparisons, where $n$ is the number of candidate nodes.
> - Second, we employ an LLM as a general classifier to evaluate the plausibility of proposed events, entities, and properties within the graph. This method ensures that each graph component adheres to the constraints of the simulation engine.
>    - Details: Our edge construction leverages an LLM as a general classifier to evaluate the plausibility of candidate connections in the context of the source node. For event-to-event edges, this process identifies causal relationships. For entity-to-event and property-to-event edges, it verifies whether the entities (e.g., an ambulance) and their properties (e.g., a siren) are supported by the simulation engine. These entities and properties are defined in the asset database (see Appendix 1.1), ensuring compatibility with the simulation of the specified event. We conducted an ablation study (more details in Appendix A.3) to evaluate the accuracy of each edge type.
>
> > Other domains not extensively explored
>
> We appreciate the reviewer’s comment regarding the exploration of other domains. Most existing works tend to focus either on driving scenarios [A1, A2, A3] or manipulation tasks [A4, A5, A6], often neglecting cross-domain exploration. In our work, we chose driving and manipulation as they represent two important fields of interest to the robotic and simulation communities. We would welcome the reviewer’s suggestions on additional domains that could benefit from our approach or serve as compelling directions for future work.
>
>
> Citations:
>
> A1. X. Yang et al., “DriveArena: A Closed-loop Generative Simulation Platform for Autonomous Driving,” arXiv.org, 2024. https://arxiv.org/abs/2408.00415.
>
> A2. Zhang, Jiawei, et al. “ChatScene: Knowledge-Enabled Safety-Critical Scenario Generation for Autonomous Vehicles.” ArXiv.org, 2024, arxiv.org/abs/2405.14062.
>
> A3. Hu, Anthony, et al. “GAIA-1: A Generative World Model for Autonomous Driving.” ArXiv.org, 2023, arxiv.org/abs/2309.17080.
>
> A4. Wang, Yufei, et al. “RoboGen: Towards Unleashing Infinite Data for Automated Robot Learning via Generative Simulation.” ArXiv.org, 2 Nov. 2023, arxiv.org/abs/2311.01455.
>
> A5. Wang, Lirui, et al. “GenSim: Generating Robotic Simulation Tasks via Large Language Models.” ArXiv.org, 2023, arxiv.org/abs/2310.01361.
>
> A6. Hua, Pu, et al. “GenSim2: Scaling Robot Data Generation with Multi-Modal and Reasoning LLMs.” ArXiv.org, 2024, arxiv.org/abs/2410.03645.

---

> > ### Author Response · Authors · 2024-11-29
> > **Please let us know your feedback**
> >
> > Dear reviewer q9yg,
> >
> > We sincerely appreciate the time you dedicated to reviewing our paper and offering constructive suggestions.
> >
> > In response to your feedback, we have incorporated additional experiments and expanded the discussion in the revised manuscript, with changes highlighted in red. As the rebuttal deadline approaches, could you kindly let us know if you have any remaining concerns?
> >
> > We look forward to hearing from you.
> >
> > Best regards,
> >
> > Authors.

---

> > > ### Author Response · Authors · 2024-12-01
> > > **2 Days Until Rebuttal Deadline – Please give us feedback**
> > >
> > > Dear Reviewer q9yg,
> > >
> > > We wanted to follow up on our earlier response to your feedback on our manuscript.
> > >
> > > As previously mentioned, we have incorporated additional experiments and expanded the discussion in the revised manuscript, with all changes highlighted in red. With the rebuttal deadline in 2 days, we kindly ask if there are any remaining concerns or points you would like us to address.
> > >
> > > We look forward to hearing from you. Thank you once again for your time and consideration.
> > >
> > > Best regards,
> > > Authors

---

> ### Comment · Reviewer_q9yg · 2024-12-02
> **Response to Authors**
>
> Thank the authors for replying to my questions. All my concerns are clear now clear. As for additional domains, I suggest the authors to add more human-robot interaction scenarios. I increased my rate to 6.

---

> ### Author Response · Authors · 2024-12-02
> **Thank you for raising the score**
>
> Dear Reviewer q9yg,
>
> Thank you very much for your response and for increasing your rating. We appreciate your suggestion regarding human-robot interaction scenarios and would like to highlight some examples from our work that align with this direction. In the driving domain, we have simulated scenarios such as "yielding to an emergency vehicle" and "picking up a passenger on the sidewalk," which involve human-robot interactions that prior methods [A1, A2, A3] were unable to address. Similarly, in the manipulation domain, we have simulated scenarios like "retrieving food from a fridge for a guest," "storing detergent out of children's reach," and "letting a guest out the door." We look forward to further advancing these directions in future work.
>
> Best,
>
> Authors
>
> ---
>
> A1. T. Wang, E. Xie, R. Chu, Z. Li, and P. Luo, “DriveCoT: Integrating Chain-of-Thought Reasoning with End-to-End Driving,” arXiv.org, 2024. https://arxiv.org/abs/2403.16996
>
> A2. C. Sima et al., “DriveLM: Driving with Graph Visual Question Answering,” arXiv.org, Dec. 21, 2023. https://arxiv.org/abs/2312.14150
>
> A3. Zhang, Jiawei, et al. “ChatScene: Knowledge-Enabled Safety-Critical Scenario Generation for Autonomous Vehicles.” ArXiv.org, 2024, arxiv.org/abs/2405.14062.

---

### Author Response · Authors · 2024-11-29
**General Response**

We thank all reviewers for their thoughtful and constructive feedback. We are encouraged to hear the reviewers,

- find ReGen’s core functionality appealing (Reviewer hjAh) as it addresses an important research problem (Reviewers HrM2, q9yg) that will improve the robustness and safety of policies under corner cases (Reviewers q9yg, HrM2, x8VP)

- great experiments to show strong capability (Reviewer hjAh) and comprehensive evaluation against multiple baselines (Reviewer x8VP) and thorough implementation across multiple domains (Reviewers x8VP, q9yg)

In response to the feedback, we have provided individual responses to address each reviewer's remaining concerns. Additionally, we have updated the manuscript, with all changes highlighted in red to enhance clarity and address any missing details. Below, we summarize the added experiments and revisions to the paper.

- Added extensive information on prompts, examples, and code, including thorough demonstrations of the entire process for constructing the simulation environment (Reviewers HrM2, x8VP, hjAh)

- Revised the method description for greater clarity (Reviewers x8VP, HrM2, hjAh), incorporating details such as model versions and hyperparameter settings (Reviewer HrM2).

- Conducted a comprehensive analysis to enhance the depth of discussion, addressing why our method outperforms the baseline (Reviewer HrM2), its handling of failure modes (Reviewer x8VP), and its novel contributions (Reviewer hjAh).

- Performed ablation experiments to clearly demonstrate method’s novelty and contribution (Reviewer hjAh)

For more details, please refer to our individual responses. We extend our gratitude to all reviewers for their time and valuable feedback. Please do not hesitate to share any further comments or suggestions.

---

### Meta-Review · Area_Chair_PWRP · 2024-12-17

**Metareview:**

This paper proposes a method that generates simulations based on trajectories and behaviors, i.e., ReGen infers the underlying scenarios and environments that could have caused the behavior. It is an interesting idea in addition to the existing generative simulation works. The initial version of this paper missed several core parts to analyze its core contribution. After rebuttal, most of the issues are resolved. However, the lack of rigorous evaluation makes this a very borderline paper. Considering the clarity of the presentation, tremendous improvement during rebuttal, and the contribution of the idea itself, I would recommend this paper for acceptance.

**Additional Comments On Reviewer Discussion:**

The authors addressed most of the concerns. However, some concerns remain for one reviewer.

---

### Decision · Program_Chairs · 2025-01-22

Accept (Poster)